# ADVERSARIALLY ROBUST NEURAL LYAPUNOV CONTROL

## ABSTRACT

State-of-the-art learning-based stability control methods for nonlinear robotic systems suffer from the issue of reality gap, which stems from discrepancy of the system dynamics between training and target (test) environments. To mitigate this gap, we propose an adversarially robust neural Lyapunov control (ARNLC) method to improve the robustness and generalization capabilities for Lyapunov theory-based stability control. Specifically, inspired by adversarial learning, we introduce an adversary to simulate the dynamics discrepancy, which is learned through deep reinforcement learning to generate the worst-case perturbations during the controller's training. By alternatively updating the controller to minimize the perturbed Lyapunov risk and the adversary to deviate the controller from its objective, the learned control policy enjoys a theoretical guarantee of stability. Empirical evaluations on five stability control tasks with the uniform and worst-case perturbations demonstrate that ARNLC not only accelerates the convergence to asymptotic stability, but can generalize better in the entire perturbation space.

## 1 INTRODUCTION

Designing a stable and robust controller to stabilize nonlinear dynamical systems has long been a challenge. Lyapunov stability theory performs a fairly significant role in the controller design for stability control of robotic systems (Uddin et al., 2021; Sharma & Kumar, 2020; Liu et al., 2020b; Norouzi et al., 2020; Pal et al., 2020). However, many previous approaches are restricted to the polynomial approximation of system dynamics (Kwakernaak & Sivan, 1969; Parrilo, 2000), and suffer from sensitivity issues when searching for the Lyapunov functions (Löfberg, 2009). Recently, by leveraging deep learning-based methods, some works have successfully incorporated the Lyapunov stability theory with the powerful expressiveness of neural networks and convenience of gradient descent for network learning (Chang et al., 2019; Abate et al., 2020; Mehrjou et al., 2021; Dawson et al., 2022). One outstanding method among them is the neural Lyapunov control (NLC) (Chang et al., 2019), where both the Lyapunov function and controller policy are approximated by neural networks. In NLC, the networks are trained by minimizing a Lyapunov risk stemmed from the Lyapunov stability theorem. Nevertheless, most existing learning-based controller are trained without any distinction between the training and test environments (Cobbe et al., 2019; Witty et al., 2021). Since the training simulator cannot perfectly model the target environment for testing, a reality gap will incur inevitably by such a modelling error (i.e., discrepancy of system dynamics), which degrades the performance of controller at the actual deployment. Hence, learning-based controller needs to consider the uncertainty of physical parameters (or external forces) that may cause the modelling error (Liu et al., 2020a; Garg & Panagou, 2021; Islam et al., 2015; Zhao et al., 2020). Motivated by this, we focus in this paper on addressing the challenging problem of learning a controller to stabilize the nonlinear dynamical system in face of such a modelling error.

Over the years, several approaches have already been proposed to alleviate the controller's performance degradation incurred by modelling errors. The majority of them is built upon another splendid learning-based control method: deep reinforcement learning (RL) (Sutton & Barto, 2018; Schulman et al., 2017b). These deep RL-based control methods treat the modeling error as an extra disturbance to the system (Başar & Bernhard, 2008), and have achieved a great success in controlling (Pinto et al., 2017; Tessler et al., 2019; Zhang et al., 2020; 2021; Mankowitz et al., 2020). For example, in robust adversarial reinforcement learning (RARL), the policy learning is formulated as a zero-sum game between the controller and an adversary that generates disturbances to the system,

where the learned controller is proved to have improved capability of robustness and generalization. Since RL methods train policies by maximizing the sum of expected rewards that the agent obtains during the interaction with environment, its performance depends greatly on the manually designed reward function while the learned policy is sensitive to the preset control interval (Tallec et al., 2019; Park et al., 2021). Hence, RL is prone to fail in the control tasks with a relatively small control interval, as will later be verified in our experiments. While our aim is to find the control policy that can enable a stable control, which is also robust to the choice of control intervals.

In this paper, we present a novel method that can automatically learn robust control policies with a provable guarantee of stability. Specifically, we formulate a perturbed Lyapunov risk for learning a controller in the dynamical system, which is imposed with the adversary's perturbations in a certain range. To train the controller policy to resist to the worst-case perturbations within that range, we formulate the learning of adversary as a Markov decision process (MDP), and train the adversary policy by proximal policy optimization (PPO). In the case of known system dynamics, the action space in the MDP can be the range of external forces or space of physical parameters, which causes the modelling error. More practically for the unknown dynamics, the original NLC no longer works since update of the networks is infeasible without prior knowledge of the system dynamics. We therefore train an environment model by sampling from the system, while the adversary's action is set as the offset to the output of this environment model. We further formulate an adversarially robust controller learning problem, which is approximately solved by alternatively updating the controller policy with Lyapunov methods and the adversary policy by PPO. Our contributions can be summarized as follows.

- We propose a perturbed Lyapunov risk for learning the control policy under perturbations.
- We formulate an optimization problem for adversarially robust controller learning, to learn a policy in face of the worst-case perturbations that are imposed by the RL-trained adversary.
- We propose an adversarially robust neural Lyapunov control (ARNLC) approach to approximately solve this problem, and demonstrate its performance on several stability control tasks.

## 2 RELATED WORK

**Adversarial training.** The idea of viewing the gap between training and test scenarios as an extra disturbance of the system was first proposed in Morimoto & Doya (2005), with the problem formulated as finding a min-max solution of the value function that takes the perturbations into account. Inspired by Morimoto & Doya (2005), Pinto et al. (2017) propose the robust adversarial reinforcement learning (RARL), where an adversary is learned simultaneously to prevent the agent from accomplishing its goal, while the agent's policy and the adversary policy are trained alternately. Zhang et al. (2020) propose robust reinforcement learning based on perturbations on state observations, introducing an adversary to apply disturbances on the state observations of the agent. Tessler et al. (2019) focus on a scenario where the agent attempts to perform an action, which behaves differently from expected due to disturbances. All of the above literature mainly studies training the adversary for RL settings, while our focus in this work is on introducing adversarial training to the Lyapunov stability control.

**Neural Lyapunov stability control.** Chang et al. (2019) propose the neural Lyapunov control, which uses neural networks to learn both the control and Lyapunov functions for nonlinear dynamical systems based on the Lyapunov stability theory. Saha et al. (2021) learn a control law that stabilizes an unknown nonlinear dynamic system. However, it needs to design a Lyapunov function manually. Dawson et al. (2022) also propose an approach for learning the robust nonlinear controller based on the robust convex optimization and Lyapunov theory, achieving generalization beyond system parameters seen during the training process. However, this approach only considers the control-affine dynamical systems, not the more general nonlinear ones. In this work, we focus on improving the robustness and generalization for control policies of nonlinear dynamical systems.

**Robust model predictive control (Robust MPC).** Robust MPC is another research branch to deal with the uncertainty in physical parameters (Sun et al., 2018; Hu & Ding, 2019; Köhler et al., 2021). It looks for the optimal feedback law among all the feasible feedback laws within a given finite horizon, in terms of a given control performance criterion at every sampling instant (Houska & Villanueva, 2019). However, it is usually restricted to the additive disturbances (Löfberg, 2003) and is computationally expensive (Bemporad & Morari, 1998).

## 3 PRELIMINARIES AND BACKGROUND

We consider a continuous-time, time-invariant nonlinear dynamical system of the form:

$$\dot{x}_t = f(x_t, a_t), \tag{1}$$

where $x_t \in \mathcal{X} \subseteq \mathbb{R}^n$ is the state and $a_t \in \mathcal{A} \subseteq \mathbb{R}^m$ is the control input at time $t$, respectively, and $\dot{x}_t$ denotes the first-order time derivative of $x_t$. The system is feedback controlled by a policy function: $a_t = \pi(x_t)$. We aim to stabilize this system at an equilibrium point $x = 0 \in \mathcal{X}$, by finding a policy to build a closed-loop controlled dynamical system $\dot{x}_t = f_\pi(x_t)$ with $f_\pi(0) = 0$, such that the equilibrium point $x = 0$ achieves asymptotic stability as defined below.

**Definition 1** (**Asymptotic stability in the sense of Lyapunov** (Lavretsky & Wise, 2013)). The equilibrium point $x = 0$ of $f_\pi$ is stable in the sense of Lyapunov if $\forall \varepsilon > 0$, $\forall t_0 > 0$, there exists $\delta(\varepsilon) > 0$ such that if $\|x(t_0)\| \leq \delta(\varepsilon)$ then $\|x(t)\| \leq \varepsilon$ for all $t \geq t_0$. The equilibrium point $x = 0$ of $f_\pi$ is asymptotically stable if it is stable and there exists a positive constant $c = c(t_0)$ such that $x(t) \to 0$ as $t \to \infty$, for all $\|x(t_0)\| \leq c$.

### 3.1 STABILITY GUARANTEE WITH LYAPUNOV FUNCTIONS

Lyapunov stability theory provides an elegant way to guarantee the stability, as follows.

**Theorem 1** (**Lyapunov stability theorem** (Lavretsky & Wise, 2013)). *Suppose $f_\pi : \mathcal{X} \to \mathbb{R}^n$ is locally Lipschitz in $\mathcal{X} \subseteq \mathbb{R}^n$. For a continuous-time controlled dynamical system $f_\pi$, if there exists a continuous function $V : \mathcal{X} \to \mathbb{R}$ such that*

$$V(0) = 0; \text{ and } V(x) > 0, \forall x \in \mathcal{X} - \{0\}; \text{ and } \dot{V}(x) < 0; \tag{2}$$

*then the system is asymptotically stable at $x = 0$, where $V$ is called a Lyapunov function.*

The time derivative of $V(x)$ can be derived as $\dot{V}(x) = \sum_{i=1}^{n} \frac{\partial V}{\partial x_i} \dot{x}_i = \sum_{i=1}^{n} \frac{\partial V}{\partial x_i} [f_\pi]_i(x)$, which depends on both $V(x)$ and the controlled dynamics $f_\pi$. Theorem 1 then states that the trajectories of the system's state will eventually reach the equilibrium $x = 0$, if we can design a control policy $\pi$ such that the Lyapunov function $V(x)$ exists and satisfies the conditions in Eq. (2).

### 3.2 NEURAL LYAPUNOV CONTROL

Neural Lyapunov control (NLC) (Chang et al., 2019) leverages neural networks to approximate both the control policy $\pi$ and Lyapunov function $V_\theta(x)$, which are parameterized by $\theta^\pi$ and $\theta$, respectively. The network parameters $\pi$ and $V_\theta(x)$ are learned by minimizing the following Lyapunov risk:

$$L_\rho(\theta, \theta^\pi) = \mathbb{E}_{x \sim \rho(\mathcal{X})} \left( \max(0, -V_\theta(x)) + \max(0, \dot{V}_\theta(x)) + V_\theta^2(0) \right), \tag{3}$$

where $x$ is a random variable following a uniformly random distribution $\rho$ over the state space $\mathcal{X}$. In its physical meaning, this Lyapunov risk quantifies the degree of violation of the Lyapunov conditions in Eq. (2) over the state space $\mathcal{X}$, given a certain policy and Lyapunov function.

## 4 ADVERSARIALLY ROBUST NEURAL LYAPUNOV CONTROL

In this paper, we aim to narrow down the reality gap incurred by the modelling discrepancy in the system dynamics $f$ between training and test environments, by learning a policy of controller $\mu$ that is better stabilizing the system (i.e., achieving the asymptotic stability faster) and more robust (i.e., resisting to variations of the system dynamics). Specifically, we consider modeling such a discrepancy by an adversary, with the system function given by:

$$\dot{x}_t = f(x_t, a_t^\mu, a_t^\nu), \tag{4}$$

where $a_t^\mu \in \mathcal{A}$ and $a_t^\nu \in \mathcal{A}_{adv}$ are the controller's action and adversary's action at time $t$ following their policies $\pi_\mu$ and $\pi_\nu$, respectively, while the rest notations follow the definition in Eq. (1). In view of the controller, $a_t^\nu$ imposes a variation to and makes the dynamics $f$ time-varying, which can be rewritten as $\dot{x}_t = f_{\pi_\nu(x_t)}(x_t, a_t^\mu)$. Hence, introducing the adversary $\nu$ imposes a time-varying modelling error to Eq. (1) during training of the controller. A practical example is where

the controller is applied to manipulate locomotion of a robot outdoor, while the adversary may be the weather that produces unpredictable wind or rain to disturb this controller. Note that the system dynamics in view of the controller reduces to Eq. (1) when $a_t^\nu = \pi_\nu(x_t) = 0$ for any time $t$. In this section, we propose the adversarially robust neural Lyapunov control (ARNLC) method to train a policy $\pi_\mu$ for the controller $\mu$, such that the system governed by Eq. (4) is stabilized in face of the adversary $\nu$.

## 4.1 PERTURBED CONTROLLER LEARNING

In our adversarial control setting, both the controller $\mu$ and adversary $\nu$ observe the system state $x_t$ at time $t$, and then take actions $a_t^\mu \sim \pi_\mu(x_t)$ and $a_t^\nu \sim \pi_\nu(x_t)$. After that, the system evolves according to Eq. (4). Here, we utilize neural networks to learn the controller policy $\pi_\mu(x_t)$, adversary policy $\pi_\nu(x_t)$ and candidate Lyapunov function $V_\theta(x)$, as parameterized by $\theta^\mu$, $\theta^\nu$ and $\theta$, respectively. Our objective is to leverage the Lyapunov stability theory to find a controller policy $\pi_\mu$ that can achieve the stability of system in the presence of a certain adversary. Namely, the resulting closed-loop controlled dynamical system $\dot{x}_t = f_{\pi_\mu, \pi_\nu}(x_t)$ is asymptotically stable at the equilibrium point $x = 0$. Motivated by this, our proposed ARNLC seeks to minimize the following perturbed Lyapunov risk w.r.t. $\theta$ and $\theta_\mu$, to update the controller policy together with the candidate Lyapunov function.

**Definition 2** (**Perturbed Lyapunov risk for controller**). We consider a candidate Lyapunov function $V_\theta$ for a continuous-time dynamical system in Eq. (4). In the presence of an adversary policy $\pi_\nu$ parameterized by $\theta_\nu$, the perturbed Lyapunov risk for the controller $\mu$ is defined by:

$$L_\rho(\theta, \theta^\mu, \theta^\nu) = \mathbb{E}_{x \sim \rho(\mathcal{X})}\bigg( \max(-V_\theta(x), 0) + \max(0, \dot{V}_\theta(x)) + V_\theta^2(0) \bigg), \qquad (5)$$

where $\rho(\mathcal{X})$ is the state distribution, and $\dot{V}_\theta(x) = \sum_{i=1}^n \frac{\partial V_\theta}{\partial x_i}[f_{\pi_\mu, \pi_\nu}]_i(x)$.

The learning of $\pi_\mu$ and $V_\theta$ can then be formulated as the following optimization problem:

$$\min_{\theta, \theta^\mu} L_\rho(\theta, \theta^\mu, \theta^\nu), \quad \text{s.t.} \ \dot{x} = f(x, a^\mu, a^\nu), a^\mu \sim \pi_\mu, a^\nu \sim \pi_\nu. \qquad (6)$$

This perturbed Lyapunov risk $L_\rho$ differs from the conventional Lyapunov risk in that the time derivative $\dot{V}_\theta(x)$ now depends on $f_{\pi_\mu, \pi_\nu}$ instead of $f_\pi$, which makes the closed-loop dynamical system time-varying for the controller. In practice, we use the following empirically perturbed Lyapunov risk, which is an unbiased estimator of Eq. (6):

$$L_{n,\rho}(\theta, \theta^\mu, \theta^\nu) = \frac{1}{N} \sum_{i=1}^N \bigg( \max(-V_\theta(x_i), 0) + \max(0, \dot{V}_\theta(x_i)) + V_\theta^2(0) \bigg). \qquad (7)$$

However, this practical estimator cannot guarantee satisfaction of the conditions in Theorem 1 on the entire state space $\mathcal{X}$. We thus apply additionally a falsifier to constantly find the counter-examples that violate these conditions during the training process, which is a common strategy also used in NLC. Specifically, the falsifier finds counter-example states according to the following criterion:

$$V_\theta(x) \le 0 \lor \dot{V}_\theta(x) \ge 0, \ \forall x \in \mathcal{X} - \{0\}, \qquad (8)$$

which specifies the negation of conditions in Eq. (2). During the training of $\pi_\mu$ and $V_\theta$, the falsifier constantly finds counter-examples and adds them into the training dataset.

## 4.2 ADVERSARY LEARNING

Compared with the conventional Lyapunov risk in Eq. (3), the perturbed counterpart $L_\rho$ presents some new challenges to the learning of controller policy. Due to the perturbation from adversary, the dynamical system in view of the controller becomes time-varying, which prevents the learning of its policy $\pi_\mu$ from reaching the stability as will be shortly shown in experiments. Inspired by the idea of adversarial training, the proposed ARNLC leverages reinforcement learning method to train the adversary policy. The intuition behind is that if we can train a controller under the worst-case perturbation (which degrades the performance of its policy to the most) in a certain range, the controller then obtains a conservative policy that is robust to any perturbation within that same range.

We formulate training of the adversary as a discrete-time Markov decision process (MDP) with a fixed control interval, defined as the tuple $(\mathcal{X}, \mathcal{A}_{adv}, \mathcal{A}, \mathcal{P}, r, \gamma)$, where $\gamma$ is the discount factor. The adversary agent observes the system state $x \in \mathcal{X}$ at each time step and takes action $a^\nu \in \mathcal{A}_{adv}$, while the controller acts $a^\mu \in \mathcal{A}$. The system then evolves to the next state $x'$ according to transition probability $\mathcal{P}(\cdot|x, a^\mu, a^\nu)$ between time steps, and the adversary agent receives reward $r(x, a^\mu, a^\nu)$.

**Adversary's action design**. Depending on whether or not the system dynamics $f$ can be accessed, ARNLC applies different action design for the adversary agent. *i*) For **known dynamics** $f$, the action space $\mathcal{A}_{adv}$ of adversary can be range of the external disturbing force or space of the environment parameters, which changes the system dynamics in view of the controller. The adversary action can then be directly imposed on the system, which simulates the external force (e.g., strong wind) and change of environment parameters (e.g., friction coefficient) that the controller may encounter in the test environment. *ii*) For **unknown dynamics** $f$, NLC is unable to back propagate the gradients to update $\pi_\mu$ and $V_\theta$, since $f$ is required to compute $L_\rho$. Alternatively, we use supervised learning to train an environment model $M_\eta$ that is approximated by the neural network for the unknown dynamical system. Then, $\pi_\mu$ and $V_\theta$ can be updated by the gradients of $M_\eta$. Since coefficients of $M_\eta$ do not have a clear physical meaning of the environment (which are weights and biases of the network), we define actions of the adversary as the additive error to the output of $M_\eta$. However, perturbations imposed by the adversary's action may lead to an unstable training, or even the non-existence of an asymptotically stable equilibrium. Hence, we limit the adversary's action to a certain range, which can be tuned practically to balance the stable training and adversary learning.

**Adversary's reward design**. Adversary should be assigned a higher reward if it leads the system to an unstable state at each time step, which is contrary to the controller's goal. For example, in the task where we aim to design a controller to swing the pendulum into a upright position, the reward of the adversary can be set as the square of the normalized angle between the pendulum and the vertical direction. The reward functions for training the adversary in all the tasks used in the paper are shown in Table A-3 in Appendix A.3. In general, the controller's action that tends to stabilize the system will decrease the reward for the adversary, while the adversary policy that achieves a higher reward will prevent the controller from minimizing $L_\rho$.

**Transition kernel**. The transition kernel is determined by the system dynamics $f$. Given a fixed control interval $\Delta t$, it can be derived as $\mathcal{P}(\cdot|x_t, a^\mu, a^\nu) = x_t + f(x_t, a_t^\mu, a_t^\nu)\Delta t$.

Given the controller policy $\pi_\mu$, the goal of RL is to find the optimal adversary policy $\pi_\nu^*$ that maximizes the following state value function:

$$\pi_\nu^* = arg \max_{\pi_\nu} V^{\pi_\mu}(\pi_\nu) = \mathbb{E}_{a_h^\mu \sim \pi_\mu, a_h^\nu \sim \pi_\nu, x_h \sim \mathcal{P}} \left[ \sum_{h=0}^\infty \gamma^h r(x_h, a_h^\mu, a_h^\nu) \right], \qquad (9)$$

where the state value function $V^{\pi_\mu}(\pi_\nu)$ denotes the expected cumulative discounted reward starting from initial state $x_0$. Provided with an appropriate adversary's reward design, $\pi_\nu^*$ can produce the worst-case perturbation sequence, which adversarially destabilizes the system to the most extent.

### 4.3 Adversarially Robust Controller Learning

Given an adversary policy $\pi_\nu^*$ trained by RL that performs the worst-case perturbation to the controller, we formulate the adversarially robust controller learning problem as:

$$\min_{\theta, \theta^\mu} L_\rho(\theta, \theta^\mu, \theta^\nu), \quad \text{s.t. } \dot{x} = f(x, a^\mu, a^\nu), a^\mu \sim \pi_\mu, a^\nu \sim \pi_\nu^* = arg \max_{\pi_\nu} V^{\pi_\mu}(\pi_\nu). \qquad (10)$$

Note that formulations of the objective functions for neural Lyapunov-controller and RL-adversary are totally different, hence the adversarial learning problem here cannot be formulated as a two-player zero-sum game. The proposed ARNLC in Algorithm 1 uses an alternating procedure to solve this problem approximately, where we summarize it for the case of unknown system dynamics $f$. **Training environment model:** at each iteration $e$, we sample $M_1$ transitions in the environment with a random policy and update the environment model by minimizing the error between its prediction $\mathcal{M}_\eta(x, a)$ and the next state $x'$ on the transitions. **Training controller's and adversary's policies:** at each outer iteration $i$, we perform a two-stage optimization. *i*) We train the controller policy and Lyapunov function while the adversary policy is fixed based on the Lyapunov theory. We initialize a state dataset $\mathcal{S}$ by randomly sampling from $\mathcal{X}$. At each inner iteration $j_\mu$, we construct $\{x_k, a_k^\mu, a_k^\nu, x_k'\}_k$ on the state subset of size $M_3$ from $\mathcal{S}$, and update $\pi_\mu$ and $V_\theta$ by performing

---

**Algorithm 1** Adversarially robust neural Lyapunov control (ARNLC) for unknown dynamics

---

**Input:** unknown environment $\mathcal{M}$ and state space $\mathcal{X}$
**Output:** learned policies $\pi_\mu$ and $\pi_\nu$
1: **Initialize:** $\eta$ for environment model $\mathcal{M}_\eta$, $\theta^\mu$ for controller $\mu$, $\theta^\nu$ for adversary $\nu$, $\theta$ for Lyapunov function $V_\theta$, control interval $\Delta t$, uniformly random policy $\pi_r$
2: **for** $e = 1, 2, \ldots N_e$ **do**                                      ▷ train an environment model for the system
3:       Sample transitions $\{x_i, a_i, x'_i\}_{i=1}^{M_1}$ from $\mathcal{M}$ using $\pi_r$ with $\Delta t$
4:       Update $\mathcal{M}_\eta$ by minimizing $\frac{1}{M_1} \sum_{i=1}^{M_1} |\mathcal{M}_\eta(x_i, a_i) - x'_i|^2$ w.r.t. $\eta$
5: **end for**
6: **for** $i = 1, 2, \ldots N_{iter}$ **do**                                      ▷ train controller
7:       Randomly sample $M_2$ states $\mathcal{S} = \{x_k\}_{k=1}^{M_2}$ from state space $\mathcal{X}$
8:       **for** $j_\mu = 1, 2, \ldots N_\mu$ **do**
9:             $a_k^\mu = \pi_\mu(x_k)$, $a_k^\nu = \pi_\nu(x_k)$ and $x'_k = M_\eta(x_k, a_k^\mu) + a_k^\nu$ on $\{x_k\}_{k=1}^{M_3}$ sampled from $\mathcal{S}$
10:             $\pi_\mu, V_\theta \leftarrow \min_{\theta, \theta^\mu} L_\rho(\theta, \theta^\mu, \theta^\nu)$ on $\{x_k, a_k^\mu, a_k^\nu, x'_k\}_{k=1}^{M_3}$               ▷ use SGD to update
11:             Find counter-example set $\Omega$ of size $M_4$ following criterion in Eq. (8) and $\mathcal{S} \leftarrow \mathcal{S} \cup \Omega$
12:       **end for**
13:       **for** $j_\nu = 1, 2, \ldots N_\nu$ **do**                                      ▷ train adversary
14:             $\{x_h, a_h^\mu, a_h^\nu, r_h\}_{h=1}^{N_{traj}} \leftarrow$ generate$(\mathcal{M}_\eta, \pi_\mu, \pi_\nu)$
15:             $\pi_\nu \leftarrow$ policyOptimize $\left( \{x_h, a_h^\mu, a_h^\nu, r_h\}_{h=1}^{N_{traj}}, \pi_\nu \right)$
16:       **end for**
17: **end for**

---

stochastic gradient descent (SGD) w.r.t. Eq. (6). *ii*) We train the adversary policy while the controller policy is fixed. At each inner iteration $j_\nu$, we generate transitions on learned environment model $\mathcal{M}_\eta$ with the controller's and adversary's policies. We then perform the policy optimization method from RL to update $\pi_\nu$ on these generated transitions. For known system dynamics, learning of the environment model at Lines 2-5 in Algorithm 1 is not required, and the transition generation in the two stages follows the known system dynamics $f$. Due to space limit, the proposed ARNLC for known system dynamics is provided in Appendix A.2.

## 5 EXPERIMENT

We evaluate our proposed ARNLC algorithm on several control tasks, where the system dynamics are variable by external forces or perturbations on environment parameters. We compare our ARNLC with NLC (Chang et al., 2019), PNLC (perturbed NLC) in Section 4.1, RARL (Pinto et al., 2017) and Robust MPC (Löfberg, 2003; 2012). We use proximal policy optimization (PPO) (Schulman et al., 2017a) as our baseline RL algorithm for the adversary training at Line 15 in Algorithm 1. In NLC and PNLC, we build neural networks for the controller policy and Lyapunov function, which are updated by minimizing the Lyapunov risk in Eq. (3) and Perturbed Lyapunov risk in Eq. (5), respectively. A uniformly random adversary policy is applied to impose perturbations in PNLC. In ARNLC and RARL, we build a neural network for the adversary policy and update it with PPO. In ARNLC, the controller policy is optimized together with the empirically perturbed Lyapunov function in Eq. (5), where the controller policy is learned by PPO with the negative adversary reward. Since the RL-based adversary and discrete-time predictor of environment model are learned in ARNLC, we compute the difference of Lyapunov function $V(x_{t+\Delta t}) - V(x_t)$ instead of the time derivative $\dot{V}(x_t)$ (see Appendix A.1 for detailed explanation). In robust MPC, a bounded perturbation is added to the system dynamics where the controller takes actions by searching the optimal feedback among all feasible horizons generated by the perturbed function. In our experiments, we find that the computation time for Robust MPC at each control step may exceed the control interval, which would lead to the controller failure in practice. But we simply neglect this and consider its simulation performance. For detail of the experimental settings, please refer to Appendix A.3. Our experiments are designed to answer the following questions.

- Can our proposed ARNLC still achieve asymptotic stability in face of the worst-case perturbations? Will ARNLC reach the stability faster than the other baseline algorithms?
- How will the controller's performance of ARNLC degrade in the entire perturbation space?

Table 1: Types of perturbations for each task

| Task | Perturbation Type |
| --- | --- |
| Pendulum | Mass of Ball, Length of Pole, Friction Coefficient, Gravity |
| Cart Pole | Mass of Cart, Length of Pole, Mass of Pole, Gravity |
| Car Trajectory Tracking | Velocity of Car, Radius of Path |
| 2-link Pendulum | Pole1 Length, Pole1 Position of the Center of Mass |

- Will ARNLC suffer from the issues of RL methods (Tallec et al., 2019; Park et al., 2021), i.e., being sensitive to control intervals?

## 5.1 CONTROL OF PERTURBED NONLINEAR SYSTEMS

We consider four balancing tasks. 1) **Pendulum**: balance a pendulum (one end attached to ground by a joint) by applying a force to a ball at the other end. The system has two state variables, angular position $\varphi$ and angular velocity $\dot{\varphi}$ of pendulum, and one control input $a^\mu$ on the ball; 2) **Cart Pole**: control a cart (attached to a pole by a joint) by applying a force to prevent the pole from falling. State variables of the system are cart position $x$, cart velocity $\dot{x}$, pendulum angular position $\varphi$ and pendulum angular velocity $\dot{\varphi}$. Control input is force $a^\mu$ applied to the cart; 3) **Car Trajectory Tracking**: control a car to follow a target circular path. The system has two state variables: the distance error $x_e$ and angle error $\varphi_e$ between current car position and the target path. The control input is force $a^\mu$ on the car; 4) **2-link Pendulum**: control a two-joint pendulum system (two pendulums are linked by a joint and one end is linked to ground by another joint) to keep both pendulums upright. The system has four state variables: the angular position $\varphi_1$ and angular velocity $\dot{\varphi}_1$ of the first pendulum, the angular position $\varphi_2$ and angular velocity $\dot{\varphi}_2$ of the second pendulum. Two control inputs are forces $a_1^\mu$ and $a_2^\mu$ on the two joints. We evaluate different comparison algorithms on these tasks with perturbed environment parameters as shown in Table 1, under both known and unknown system dynamics settings. For example, in the pendulum task, the friction coefficient can be changed at each time step. We set two test scenarios: *i*) perturbations are randomly generated w.r.t. a uniform distribution at each control step, which are larger than training ones; *ii*) perturbations are taken w.r.t. the trained adversary policy of ARNLC.

We run the training process of learning-based algorithms, i.e., ARNLC, NLC, PNLC and RARL on these tasks until the convergence. Here, we slightly modify and improve the original NLC and PNLC to make them compatible with the setting of unknown system dynamics. We then deploy the trained policies and robust MPC in the two test scenarios. **For known system dynamics**, control curves under uniform (U) perturbations are shown in Figs. 1(a), 1(c), 1(e) and 1(g), while control curves under worst-case (W) perturbations learned by the adversary are illustrated in Figs. 1(b), 1(d), 1(f) and 1(h). **For unknown system dynamics**, control curves under uniform (U) perturbations are shown in Figs. 1(i), 1(k), 1(m) and 1(o), while control curves under worst-case (W) perturbations learned by the adversary are illustrated in Figs. 1(j), 1(l), 1(n) and 1(p). The horizontal axis is the control time, while the vertical axis is the system state. Here we set the fixed control interval to 0.01s, and state zero as the equilibrium point. We observe that by incorporating a RL-based adversary during training, our ARNLC can achieve asymptotic stability under both test scenarios in all the tasks, while it reaches the stability the fastest compared to the other baselines under both conditions of known and unknown system dynamics. Though NLC reaches the stability in some tasks, it fails to reach the equilibrium point in Car Tracking W, 2-link Pendulum U and W with known system dynamics and Car Tracking U and W, 2-link Pendulum W with unknown system dynamics. PNLC trained under uniform sampled perturbations outperforms NLC in some tasks (e.g., Pendulum and Cart Pole), but is worse in Cart Tracking U and 2-link Pendulum W and U with known system dynamics and 2-link Pendulum U with unknown system dynamics. RARL and robust MPC fail to reach the stability in the 2-link Pendulum task.

## 5.2 GENERALIZATION IN PERTURBATION SPACE

We further evaluate the generalization capability of trained policies of learning-based algorithms in the entire perturbation space. We exclude the evaluation of robust MPC here, since it requires to know system dynamics under each perturbation setting, which is an unfair comparison. Besides, we additionally evaluate ARNLC and RARL for unknown system dynamics in the **Inverted Pendulum**

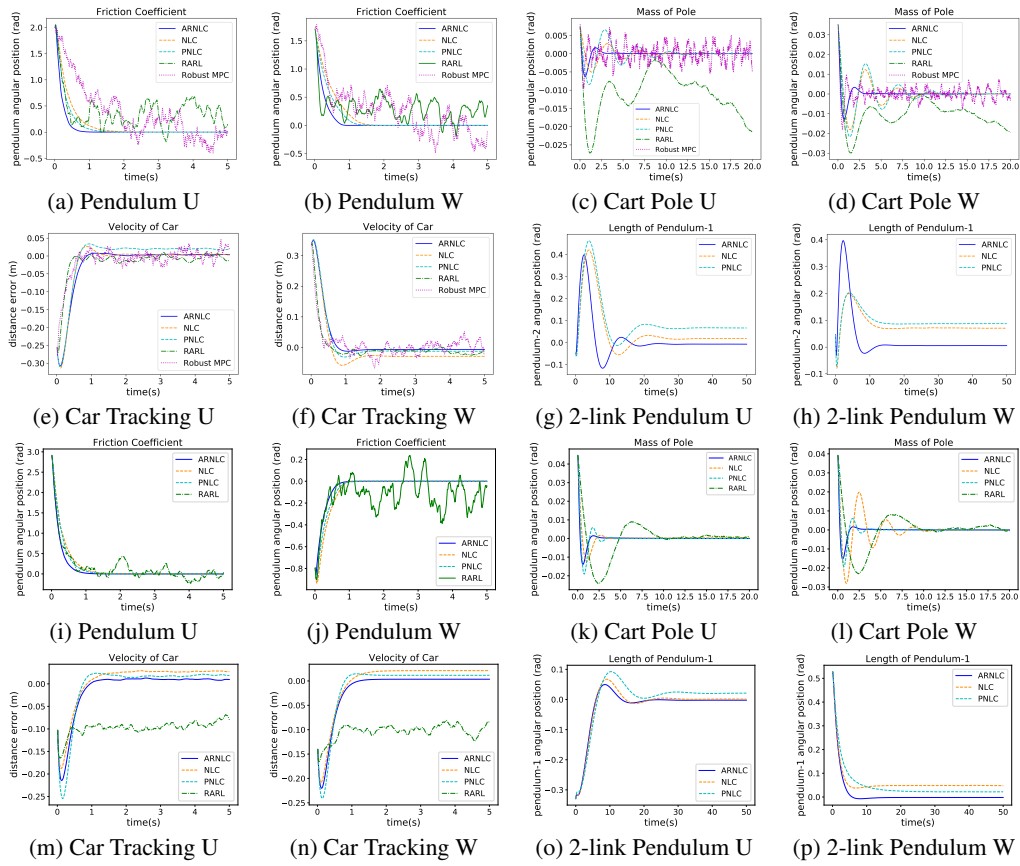

Figure 1: Control curves of Pendulum, Cart Pole, Car Tracking and 2-link Pendulum with different perturbation types under uniform perturbations (U) and learned adversary's worst-case (W) perturbations in testing. 1(a))-1(h)): known system dynamics; 1(i)-1(p): unknown system dynamics.

task provided by MuJoCo (Todorov et al., 2012), which controls a cart (attached to a pendulum) to balance the whole system and keep the pendulum upright. While NLC and PNLC are not compared, since they require known system dynamics during training and cannot be trained in the Inverted Pendulum task. We use the cumulative negative reward of adversary to evaluate the performance of controller policy in the environment with a certain perturbation, where a higher negative reward indicates the better performance of controller. The performance heatmaps of these five tasks achieved by different algorithms are shown and compared in Fig. 2 with both known and unknown system dynamics, and 3(g) with unknown system dynamics, where the performance is averaged on ten equal-length runs. We observe that ARNLC achieves the best generalization performance under both conditions of known and unknown system dynamics except for Car Tracking with unknown system dynamics. PNLC generalizes better than NLC in Pendulum and CartPole with known system dynamics, while showing similar or even worse performance in other tasks. RARL presents the worst performance in all the tasks except for Pendulum.

## 5.3 Impact of Control Intervals

Eventually, we evaluate the impact of different control intervals to our ARNLC, which are set to 0.01s, 0.1s, 0.005s and 0.001s, respectively. The resulting control curves obtained for Pendulum are shown in Figs. 1(a)-1(b) and 3(a)-3(f). We observe that ARNLC and other Lyapunov-based baselines can achieve asymptotic stability with all different control intervals, while RARL is sensitive to the change of control intervals as also verified in (Tallec et al., 2019) and fails to reach the equilibrium state. Note that here we report the most important results in the main text, please also see Appendices A.4 and A.5 for additional results.

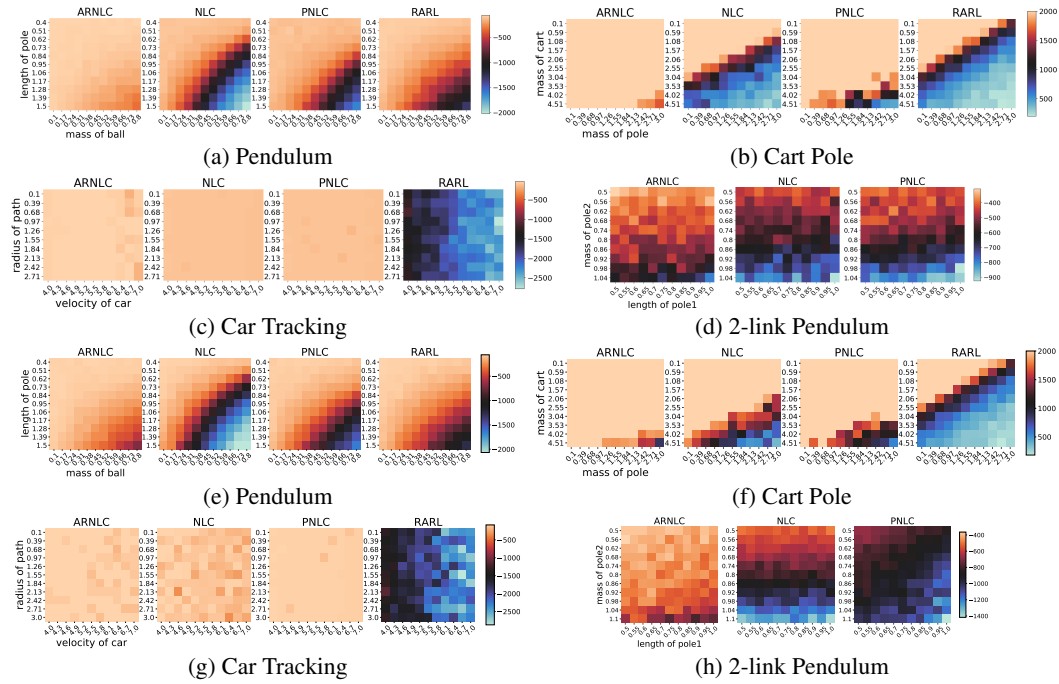

Figure 2: Heatmap of averaged cumulative negative adversary's reward with known and unknown system dynamics. In (d) and (h), RARL fails to converge. 2(a)-2(d): known system dynamics; 2(e)-2(h): unknown system dynamics.

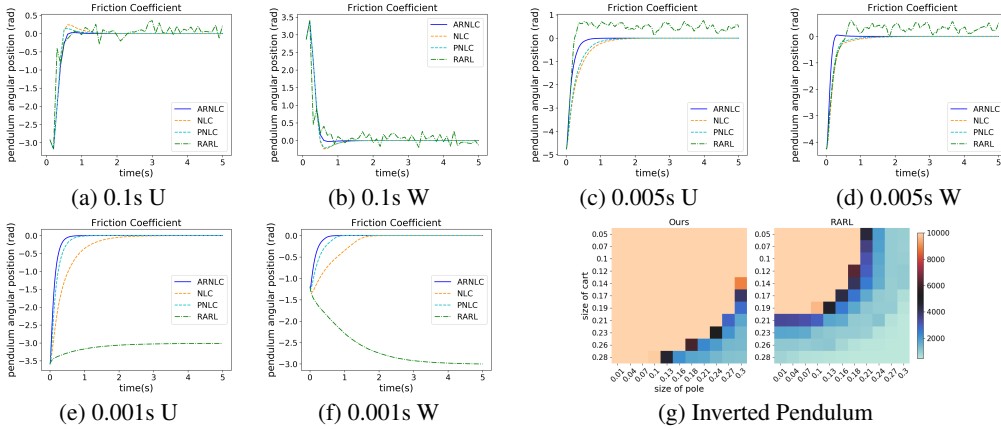

Figure 3: (a)-(f) Control curves of Pendulum with control interval set to 0.1s, 0.005s and 0.001s, respectively; (g) heatmap of averaged cumulative negative adversary's reward with unknown system dynamics.

## 6 CONCLUSIONS AND FUTURE WORK

We have proposed ARNLC to improve the robustness and generalization in stability control tasks for nonlinear dynamical systems. Specifically, we formulated a perturbed Lyapunov risk stemmed from Lyapunov theorem to jointly update the controller and candidate Lyapunov function under perturbations generated by an adversary during training, where the adversary was trained by RL method to destabilize the system. We adopted an alternative training procedure to update the controller and adversary. We have empirically evaluated ARNLC in several stability control tasks, demonstrating its robustness under different perturbations and better generalization in entire perturbation space. An exciting future research direction could be to extend our ARNLC to non-stability control tasks for the dynamical systems that are not required to achieve an equilibrium point, where the reward function for the adversary can alternatively be designed based on the imitation error, like in the imitation learning. We hope to revisit this in our future works.

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

# A   APPENDIX

## A.1   CONTROLLER LEARNING FOR DISCRETE-TIME CONTROL

We consider a discrete-time dynamical system sampled from a continuous-time dynamical system with fixed time interval $\Delta t$:

$$x'_t = x_{t+\Delta t} = x_t + \dot{x}\Delta t = f(x_t, a_t^\mu, a_t^\nu), \tag{A-1}$$

where $a_t^\mu \in \mathcal{A}$ and $a_t^\nu \in \mathcal{A}_{adv}$ are the controller's action and adversary's action at time $t \in \mathbb{N}$ following their policies $\pi_\mu$ and $\pi_\nu$, respectively; and $x'_t$ is the state at time $t + \Delta t$ (i.e., at the next sampling step).

**Theorem A-1** (**Lyapunov stability theorem for discrete-time dynamical systems**). *For a discrete-time dynamical system in Eq. (A-1), if there exists a continuous function $V \colon \mathcal{X} \to \mathbb{R}$ such that*

$$V(0) = 0; \text{ and } V(x) > 0, \forall x \in \mathcal{X} - \{0\}; \text{ and } V(x') - V(x) < 0; \tag{A-2}$$

*then the system is asymptotically stable at $x = 0$, where $V$ is called a Lyapunov function.*

For the discrete-time dynamical system, we focus on the difference of the Lyapunov function instead of the time derivative in the continuous-time case. To satisfy the Lyapunov stability theorem, we require that: *i*) the value of $V(0)$ is zero; *ii*) the value of $V(x)$ is positive; and *iii*) the value of the difference $V(x') - V(x)$ is negative.

**Definition A-1** (**Discrete-time perturbed Lyapunov risk for controller**). We consider a candidate Lyapunov function $V_\theta$ paramterized by $\theta$ for discrete-time dynamical system in Eq. (A-1). In the presence of an adversary policy $\pi_\nu$ parameterized by $\theta_\nu$, the discrete-time perturbed Lyapunov risk for the controller $\mu$ is defined by:

$$L_\rho\left(\theta, \theta^\mu, \theta^\nu\right) = \mathbb{E}_{x\sim\rho(\mathcal{X})}\left( \max(-V_\theta(x), 0) + \max(0, V_\theta(x') - V_\theta(x)) + V_\theta^2(0)\right), \tag{A-3}$$

where $\rho(\mathcal{X})$ is the state distribution.

In practice, we use the following empirically perturbed Lyapunov risk, which is an unbiased estimator of Eq. (A-3):

$$L_{n,\rho}\left(\theta, \theta^\mu, \theta^\nu\right) = \frac{1}{N}\sum_{i=1}^{N}\left( \max(-V_\theta(x_i), 0) + \max(0, V_\theta(x'_i) - V_\theta(x_i)) + V_\theta^2(0)\right). \tag{A-4}$$

We use the negations of the Lyapunov conditions in Lyapunov stability theorem to define the counter-examples, which means that if the value of $V_\theta(x)$ is non-positive or the value of the difference $V_\theta(x') - V_\theta(x)$ is non-negative, then the state $x$ is considered as a counter-example. Therefore, the criterion for discrete-time dynamical systems can be set as follows:

$$\Omega(x) = V_\theta(x) \leq 0 \vee V_\theta(x') - V_\theta(x) \geq 0, \forall x \in \mathcal{X} - \{0\}. \tag{A-5}$$

## A.2   ARNLC FOR KNOWN SYSTEM DYNAMICS

We summarize the ARNLC for known system dynamics $f$ in Algorithm A-1 of this Appendix. Its difference from ARNLC for unknown system dynamics lies in that there is no need to learn an environment model. This algorithm follows the same alternating procedure as described in Section 4.3 in the main text to train the controller's and the adversary's policies.

## A.3   DETAILS OF EXPERIMENTAL SETTINGS

The network architectures for Lyapunov function and controller policy are tuned for each task and are summarized in Table A-1. The detailed hyperparameters for adversary policy training are summarized in Table A-2. The environment specific parameters are summarized in Table A-3.

---

**Algorithm A-1** Adversarially robust neural Lyapunov control for known system dynamics

---

**Input:** known dynamical system $f$ and state space $\mathcal{X}$
**Output:** learned policies $\pi_\mu$ and $\pi_\nu$
1: **Initialize:** $\theta^\mu$ for controller $\mu$, $\theta^\nu$ for adversary $\nu$, $\theta$ for Lyapunov function $V_\theta$, control interval $\Delta t$
2: **for** $i = 1, 2, \ldots N_{iter}$ **do**
3:      Randomly sample $M_1$ states $\mathcal{S} = \{x_k\}_{k=1}^{M_1}$ from state space $\mathcal{X}$
4:      **for** $j_\mu = 1, 2, \ldots N_\mu$ **do**                                              ▷ train controller
5:          $a_k^\mu = \pi_\mu(x_k)$, $a_k^\nu = \pi_\nu(x_k)$ and $x_k' = f(x_k, a_k^\mu, a_k^\nu)$ on $\{x_k\}_{k=1}^{M_2}$ sampled from $\mathcal{S}$
6:          $\pi_\mu, V_\theta \leftarrow \min_{\theta, \theta^\mu} L_\rho(\theta, \theta^\mu, \theta^\nu)$ on $\{x_k, a_k^\mu, a_k^\nu, x_k'\}_{k=1}^{M_2}$               ▷ use SGD to update
7:          Find counter-example set $\Omega$ of size $M_3$ following criterion in Eq. (A-5) and $\mathcal{S} \leftarrow \mathcal{S} \cup \Omega$
8:      **end for**
9:      **for** $j_\nu = 1, 2, \ldots N_\nu$ **do**                                              ▷ train adversary
10:         $\{x_h, a_h^\mu, a_h^\nu, r_h\}_{h=1}^{N_{traj}} \leftarrow \text{generate}(f, \pi_\mu, \pi_\nu)$
11:         $\pi_\nu \leftarrow \text{policyOptimize}\left(\{x_h, a_h^\mu, a_h^\nu, r_h\}_{h=1}^{N_{traj}}, \pi_\nu\right)$
12:     **end for**
13: **end for**

---

Table A-1: Lyapunov function and controller policy specific settings

| Task | Network Architecture | | Training Iterations | Learning Rate | Batch Size |
|---|---|---|---|---|---|
| | Lyapunov Function | Controller Policy | | | |
| Pendulum | 2-6-1 | 2-1 | 200 | | |
| Cart Pole | 4-64-64-1 | 4-64-64-1 | 20 | | |
| Car Trajectory Tracking | 2-64-1 | 2-1 | 12 | 0.01 | 512 |
| 2-link Pendulum | 4-6-1 | 4-64-2 | 6 | | |
| Inverted Pendulum | 4-64-64-1 | 4-64-64-1 | 40 | | |

Table A-2: Adversary hyperparameters

| Parameter | Value |
|---|---|
| Optimizer | Adam |
| Learning rate for actor policy network | $3 \times 10^{-4}$ |
| Learning rate for critic network | $1 \times 10^{-3}$ |
| Discount ($\gamma$) | 0.99 |
| Clipping ratio ($\epsilon$) | 0.2 |
| Number of hidden layers (all tasks) | 2 |
| Number of hidden units per layer | 64 |
| Nonlinearity | tanh |

Table A-3: Environment specific parameters

| Task | State Space | Adversary Reward |
|---|---|---|
| Pendulum | $\|\varphi\| \leq 6, \|\dot{\varphi}\| \leq 6$ | $\varphi^2$ |
| Cart Pole | $\|x\| \leq 1, \|\dot{x}\| \leq 1, \|\varphi\| \leq 0.2, \|\dot{\varphi}\| \leq 1$ | $-1$ if $\|\varphi\| < 0.2$ |
| Car Trajectory Tracking | $\|x_e\| \leq 0.5, \|\varphi_e\| \leq 0.5$ | $\|x_e\| + \|\varphi_e\|$ |
| 2-link Pendulum | $\|\varphi_1\| \leq 0.8, \|\dot{\varphi}_1\| \leq 0.8, \|\varphi_2\| \leq 0.8, \|\dot{\varphi}_2\| \leq 0.8$ | $\|\varphi_1\| + \|\varphi_2\|$ |
| Inverted Pendulum | $\|x\| \leq 1, \|\dot{x}\| \leq 1, \|\varphi\| \leq 0.2, \|\dot{\varphi}\| \leq 1$ | $-1$ if $\|\varphi\| < 0.2$ |

### A.3.1 SYSTEM DYNAMICS

**Pendulum.** The system is shown in Fig. A-1 and the system dynamics can be described as

$$\ddot{\theta} = \frac{mgl\sin(\theta) + u - b\dot{\theta}}{ml^2}, \tag{A-6}$$

where $m = 0.15$, $g = 9.81$, $b = 0.1$ and $l = 0.5$.

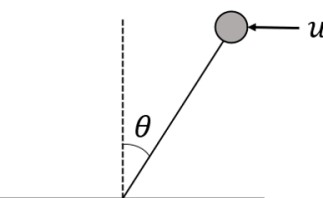

Figure A-1: Schematic diagram of the Pendulum task.

The adversary reward in this environment is set as the square of the normalized angle between the pendulum and the goal state. The actions of the adversary are set to adding an external force to the pendulum, changing the acceleration of gravity in the environment, changing the length of the pendulum, changing the coefficient of friction, and changing the mass of the ball, respectively.

**Cart Pole.** The system is shown in Fig. A-2 and the system dynamics can be described as

$$\ddot{\theta} = \frac{g \sin \theta + \cos \theta \left( \frac{-u - m_p l \dot{\theta}^2 \sin \theta}{m_c + m_p} \right)}{l \left( \frac{4}{3} - \frac{m_p \cos^2 \theta}{m_c + m_p} \right)},$$

$$\ddot{x} = \frac{u + m_p l (\dot{\theta}^2 \sin \theta - \ddot{\theta} \cos \theta)}{m_c + m_p},$$

(A-7)

where $g = 9.8$, $m_c = 1.0$, $m_p = 0.1$, $l = 0.5$.

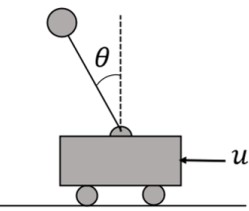

Figure A-2: Schematic diagram of the Cart Pole task.

If the angle between the pendulum and the vertical direction is less than 0.2, the adversary gets reward -1. The actions of the adversary are set to adding an external force to the cart, changing the acceleration of gravity in the environment, changing the length of the pendulum, changing the mass of the cart, and changing the mass of the ball, respectively.

**Car Trajectory Tracking.** The system is shown in Fig. A-3 and the system dynamics can be described as

$$\dot{s} = \frac{v \cos(\theta_e)}{1 - \dot{d}_e \kappa(s)},$$

$$\dot{d}_e = v \sin(\theta_e),$$

(A-8)

$$\dot{\theta}_e = \frac{v \tan(u)}{L} - \frac{v \kappa(s) \cos(\theta_e)}{1 - \dot{d}_e \kappa(s)},$$

where $v = 6$, $l = 1$.

The adversary reward in this environment is set as the sum of the absolute value of the distance error and the absolute value of the normalized angle error. The actions of the adversary are set to adding an external force to the pendulum, changing the velocity of the car, and changing the radius of the target path, respectively.

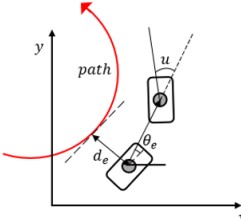

Figure A-3: Schematic diagram of the Car Trajectory Tracking task.

**2-link Pendulum.** The system is shown in Fig. A-4 and the system dynamics can be described as

$$\ddot{\theta}_1 = \frac{a_{22}[u_1 + a_{12}\dot{\theta}_2^2 \sin(\theta_2 - \theta_1) + b_1 \sin\theta_1] - a_{12}\cos(\theta_2 - \theta_1)[u_2 + a_{21}\dot{\theta}_1^2 \sin(\theta_1 - \theta_2) + b_2 \sin\theta_2]}{a_{11}a_{22} - a_{12}a_{21}\cos(\theta_1 - \theta_2)\cos(\theta_2 - \theta_1)},$$

$$\ddot{\theta}_2 = \frac{a_{21}\cos(\theta_1 - \theta_2)[u_1 + a_{12}\dot{\theta}_2^2 \sin(\theta_2 - \theta_1) + b_1 \sin\theta_1] - a_{11}[u_2 + a_21\dot{\theta}_1^2 \sin(\theta_1 - \theta_2) + b_2 \sin\theta_2]}{a_{12}a_{21}\cos(\theta_2 - \theta_1)\cos(\theta_1 - \theta_2) - a_{11}a_{22}},$$

(A-9)

where

$$a_{11} = I_1 + m_1 l_{c1}^2 + l_1^2 m_2,$$
$$a_{12} = a_{21} = m_2 l_1 l_{c2},$$
$$a_{22} = I_2 + m_2 l_{c2}^2,$$
$$b_1 = (m_1 l_{c1} + l_1 m_2)g,$$
$$b_2 = (m_2 l_{c2})g,$$

(A-10)

and $I_1 = I_2 = 1$, $m_1 = m_2 = 1$, $l_1 = l_2 = 1$, $l_{c1} = l_{c2} = 0.5$, $g = 9.8$.

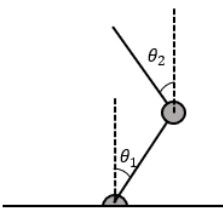

Figure A-4: Schematic diagram of the 2-link Pendulum task.

The adversary reward in this environment is set as the sum of the normalized angles between the two pendulums and the goal state. The actions of the adversary are set to adding two external forces to the pendulum, changing the length of the first pendulum, and changing the position of the center of mass of the first pendulum, respectively.

**Inverted Pendulum.** This task is provided by MuJoCo, where the goal is to control a cart (attached to a pendulum) to balance the whole system and keep the pendulum upright, as shown in Fig. A-5. Since the system dynamics of Inverted Pendulum are unknown, it is impossible to generate perturbations on concrete physical parameters at each control step. Therefore, we only conduct the generalization test in the entire perturbation space for this task. If the angle between the pendulum and the vertical direction is less than 0.2, the adversary gets reward -1. The actions of the adversary are set as the additive error to the output of the learned environment model.

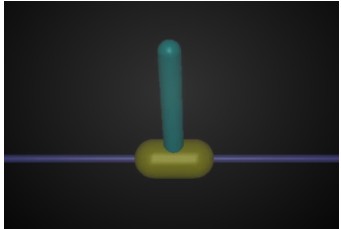

Figure A-5: Schematic diagram of the Inverted Pendulum task.

## A.4 ADDITIONAL EXPERIMENTAL RESULTS WITH KNOWN SYSTEM DYNAMICS SCENARIOS

This section reports experimental results on all the evaluation tasks under perturbation types that have not been presented in the main text.

### A.4.1 CONTROL OF PERTURBED NONLINEAR SYSTEMS

We run the training process of learning-based algorithms on Pendulum, Cart Pole, Car Trajectory Tracking, and 2-link Pendulum under different perturbation types as shown in Table 1 in the main text until convergence. Control curves under uniform (U) perturbations are shown in Fig. A-6, while control curves under worst-case (W) perturbations learned by the adversary are illustrated in Fig. A-7. We observe that our ARNLC can achieve asymptotic stability under both test scenarios in almost all the tasks, while it reaches the stability the fastest compared to the other baselines. The uniform external force perturbations in the Car Trajectory Tracking task is the only test scenario where our ARNLC performs the worst, as shown in Fig. A-6(k). Though NLC reaches the stability in some tasks, it fails to reach the equilibrium point in Car Trajectory Tracking W, 2-link Pendulum W and U. RARL and robust MPC fail to reach the stability in 2-link Pendulum.

We additionally compute the percentile of negative adversary rewards for controller policies achieved by each algorithm to further evaluate their robustness. We run each policy with 100 different initial system states in each perturbed task, and then sort the cumulative negative adversary reward of each run to obtain the $n$-th percentile. The results obtained under uniform (U) perturbations are shown in Fig. A-8, while percentile plots under worst-case (W) perturbations learned by the adversary are illustrated in Fig. A-9. In general, control policies that can gain higher rewards at the same percentile perform better. While control policies in a lower percentile have better control performance if they receive the same reward, i.e., they can gain higher rewards with more episodes. We observe that our ARNLC can receive the highest rewards under both test scenarios in all the tasks. RARL sometimes fail to reach the stability in Car Trajectory Tracking, and therefore, RARL sometimes receives extremely low rewards in this task.

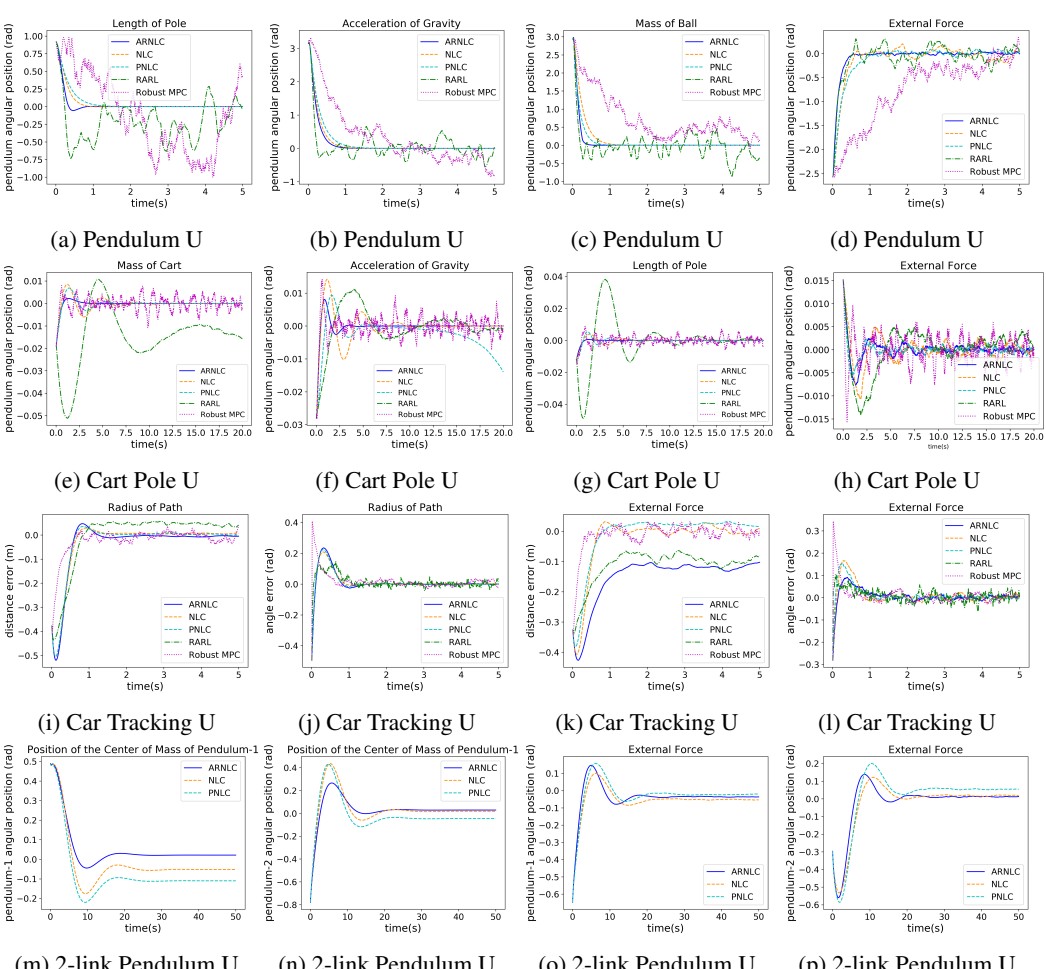

(a) Pendulum U  (b) Pendulum U  (c) Pendulum U  (d) Pendulum U

(e) Cart Pole U  (f) Cart Pole U  (g) Cart Pole U  (h) Cart Pole U

(i) Car Tracking U  (j) Car Tracking U  (k) Car Tracking U  (l) Car Tracking U

(m) 2-link Pendulum U  (n) 2-link Pendulum U  (o) 2-link Pendulum U  (p) 2-link Pendulum U

Figure A-6: Control curves of Pendulum, Cart Pole, Car Trajectory Tracking and 2-link Pendulum under uniform (U) perturbations in testing.

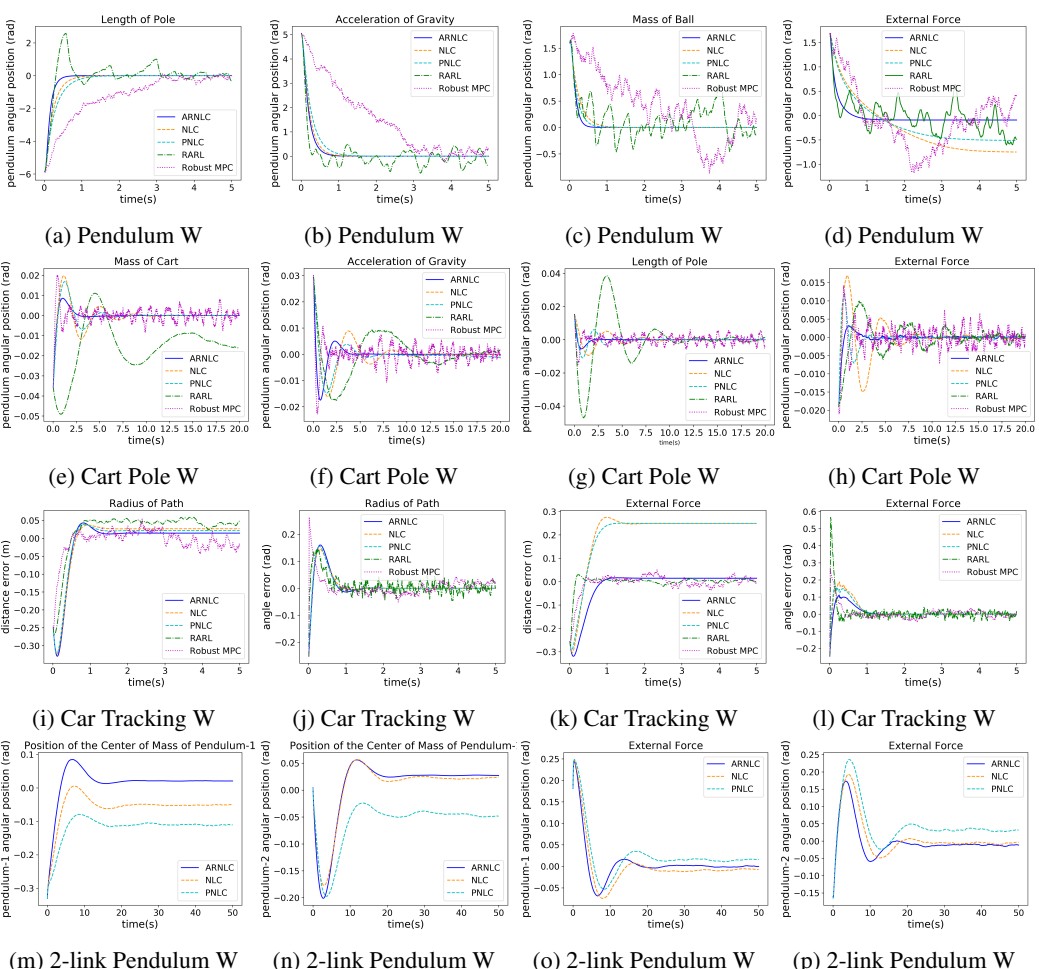

Figure A-7: Control curves of Pendulum, Cart Pole, Car Trajectory Tracking and 2-link Pendulum under learned adversary's worst-case (W) perturbations in testing.

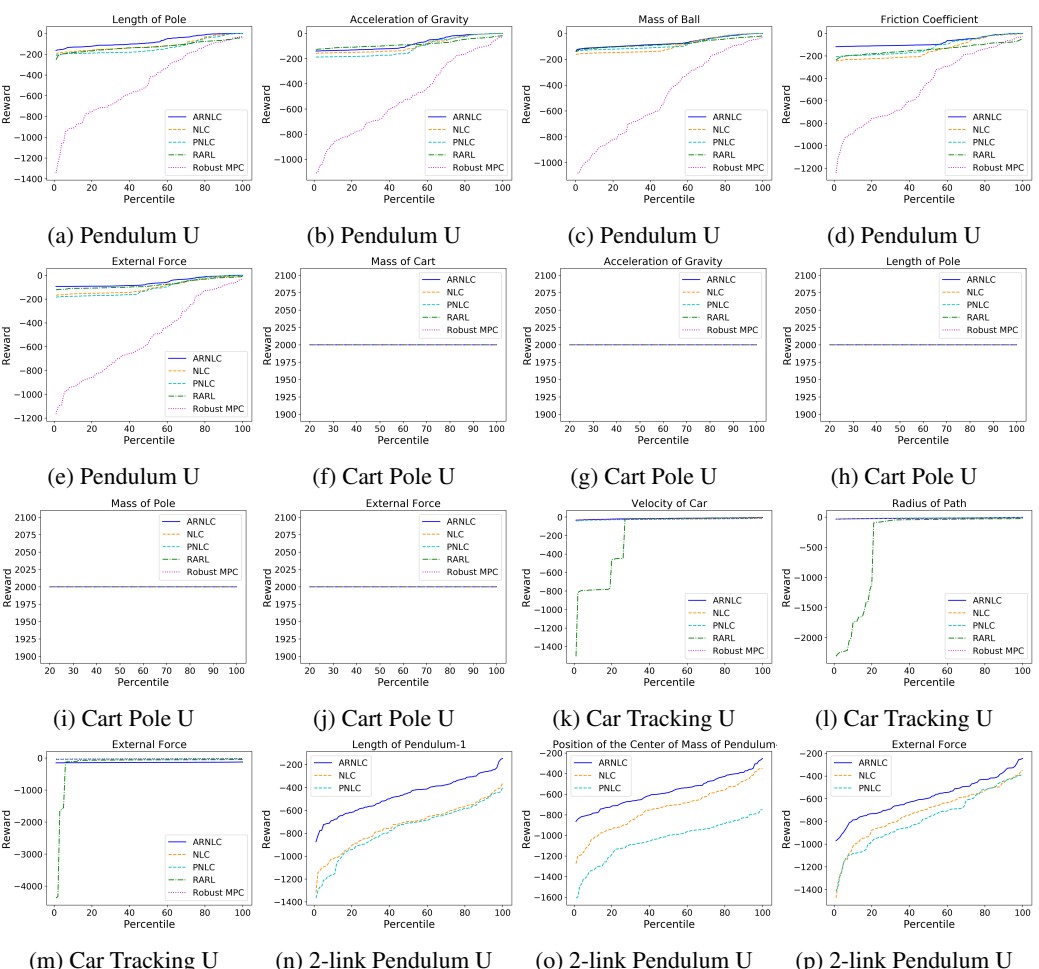

Figure A-8: Percentile plots of Pendulum, Cart Pole, Car Trajectory Tracking and 2-link Pendulum under uniform (U) perturbations in testing.

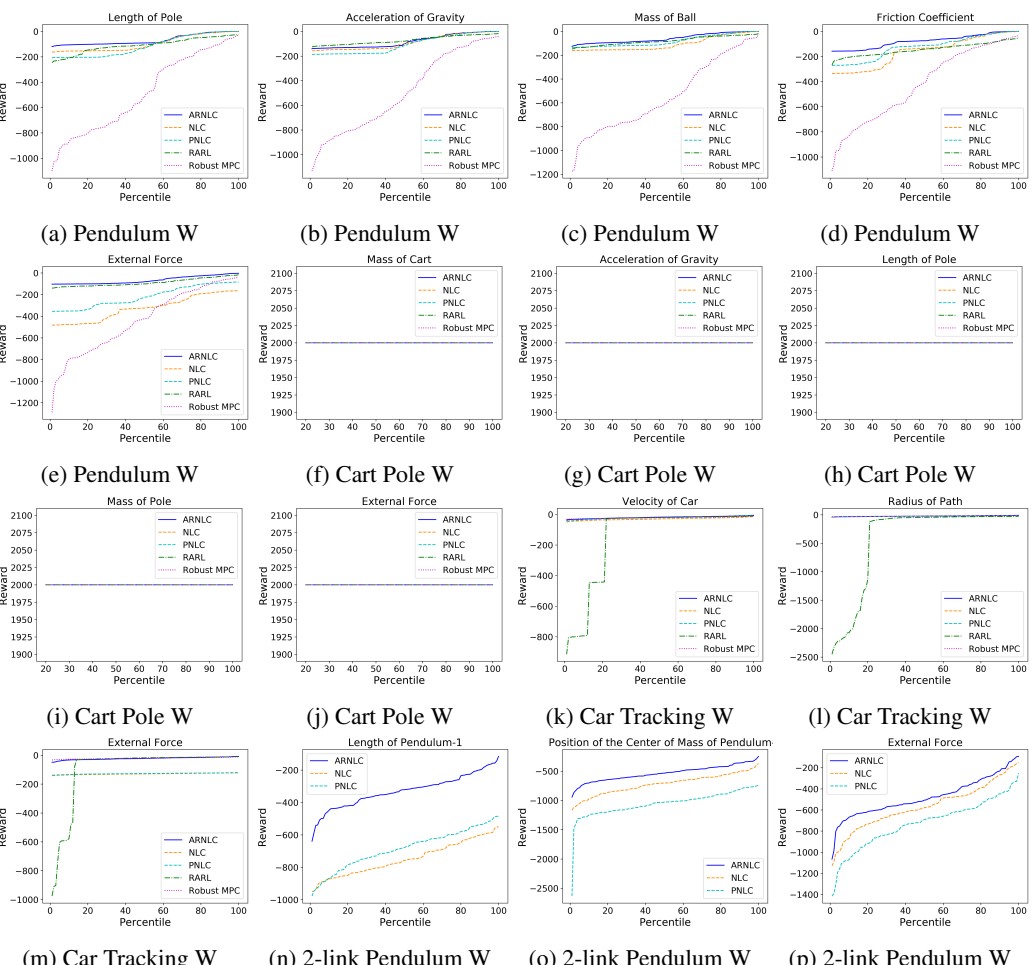

Figure A-9: Percentile plots of Pendulum, Cart Pole, Car Trajectory Tracking and 2-link Pendulum under learned adversary's worst-case (W) perturbations in testing.

### A.4.2 GENERALIZATION IN PERTURBATION SPACE

In this subsection, we evaluate the generalization capability of learning-based algorithms in the entire perturbation space generated from other combinations of perturbation types that are not presented in the main text. We observe that ARNLC achieves the best generalization performance among all the combinations of different physical parameters. PNLC generalizes better than NLC in most combinations, while showing worse performance in Figs. A-10(f) and A-10(j).

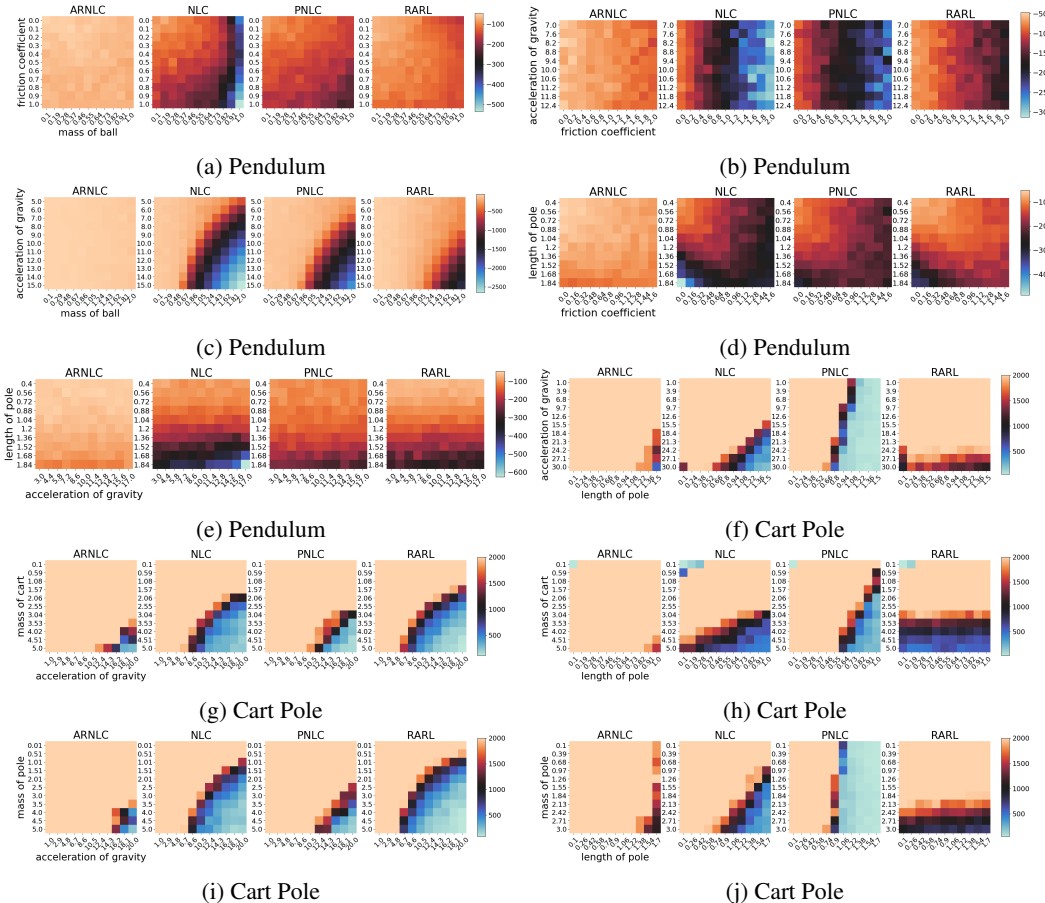

Figure A-10: Heatmap of averaged cumulative negative adversary's reward for Pendulum, Cart Pole.

### A.4.3 Impact of Control Intervals

We evaluate the impact of different control intervals (0.01s, 0.1s, 0.005s and 0.001s) to our ARNLC on Pendulum with perturbation types (Length of Pole, Mass of Ball, Friction Coefficient, Acceleration of Gravity) different from the results presented in the main text. The resulting control curves obtained for Pendulum under uniform perturbations are shown in Figs. A-6(a)-A-6(d) and Fig. A-11. And the control curves under perturbation of learned adversary are shown in Figs. A-7(a)-A-7(d) and Fig. A-12. The resulting percentile plots obtained for Pendulum under uniform perturbations are shown in Figs. A-8(a)-A-8(e) and Fig. A-13. And the percentile plots under perturbation of learned adversary are shown in Figs. A-9(a)-A-9(e) and Fig. A-14. The results demonstrate that our ARNLC is robust to different control intervals.

We further evaluate our ARNLC in the continuous-time dynamical system of Pendulum, with the system dynamics given in Eq. (A-6). Here we show that our ARNLC can achieve asymptotic stability faster than NLC in Figs. A-15(a)-A-15(d) and can receive higher rewards in Figs. A-15(e)-A-15(h). The regions of attraction are shown in Figs. A-15(i)-A-15(l), demonstrating that those estimated by our ARNLC are comparable with those estimated by neural Lyapunov Controller. This indicates that we still have the advantage of the larger region of attraction inherited from the original method after incorporating the adversarial training.

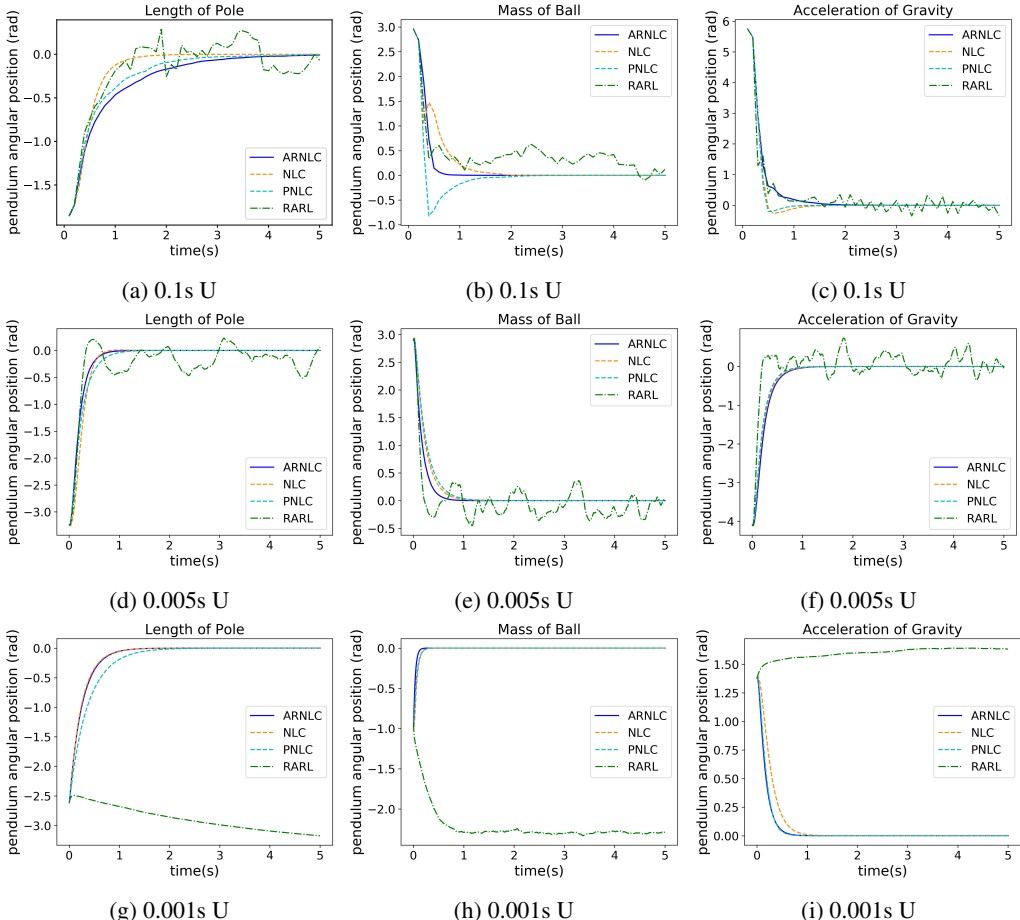

Figure A-11: Control curves of Pendulum under uniform (U) perturbations in testing with control interval set to 0.1s, 0.005s and 0.001s, respectively.

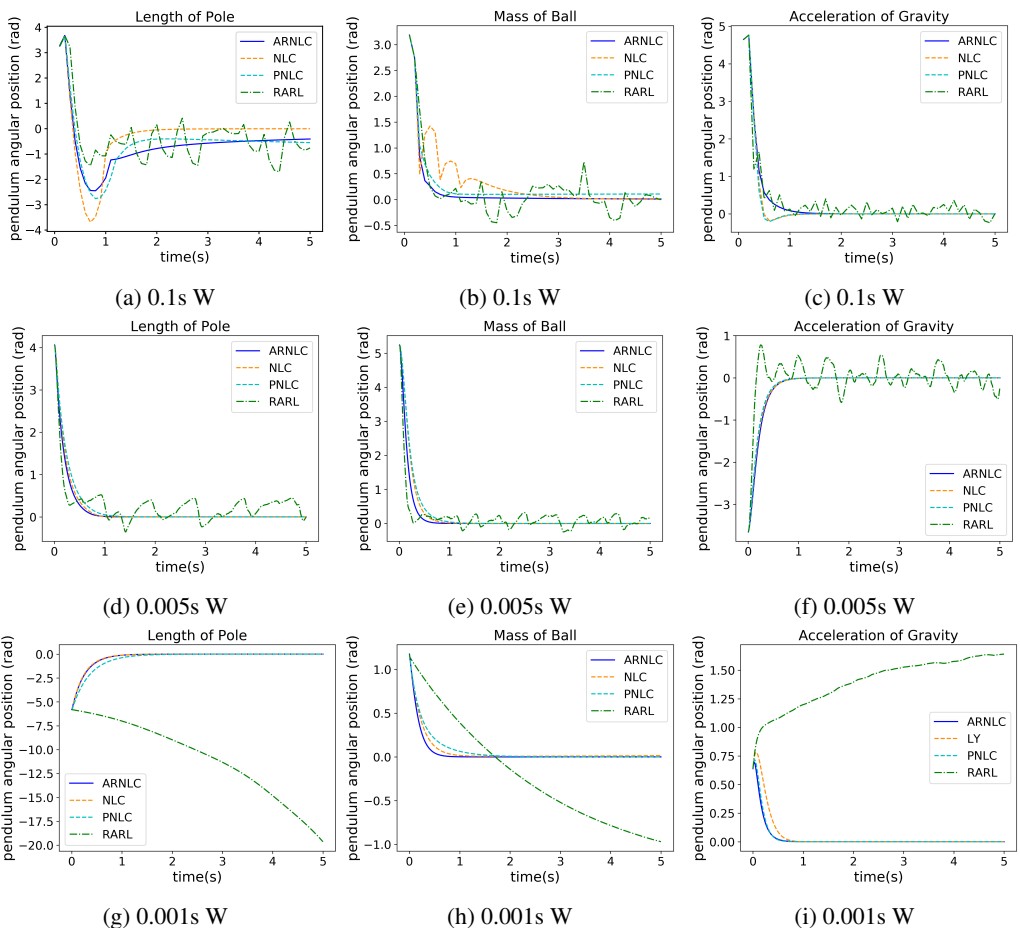

(a) 0.1s W      (b) 0.1s W      (c) 0.1s W

(d) 0.005s W      (e) 0.005s W      (f) 0.005s W

(g) 0.001s W      (h) 0.001s W      (i) 0.001s W

Figure A-12: Control curves of Pendulum under learned adversary's worst-case (W) perturbations in testing with control interval set to 0.1s, 0.005s and 0.001s, respectively.

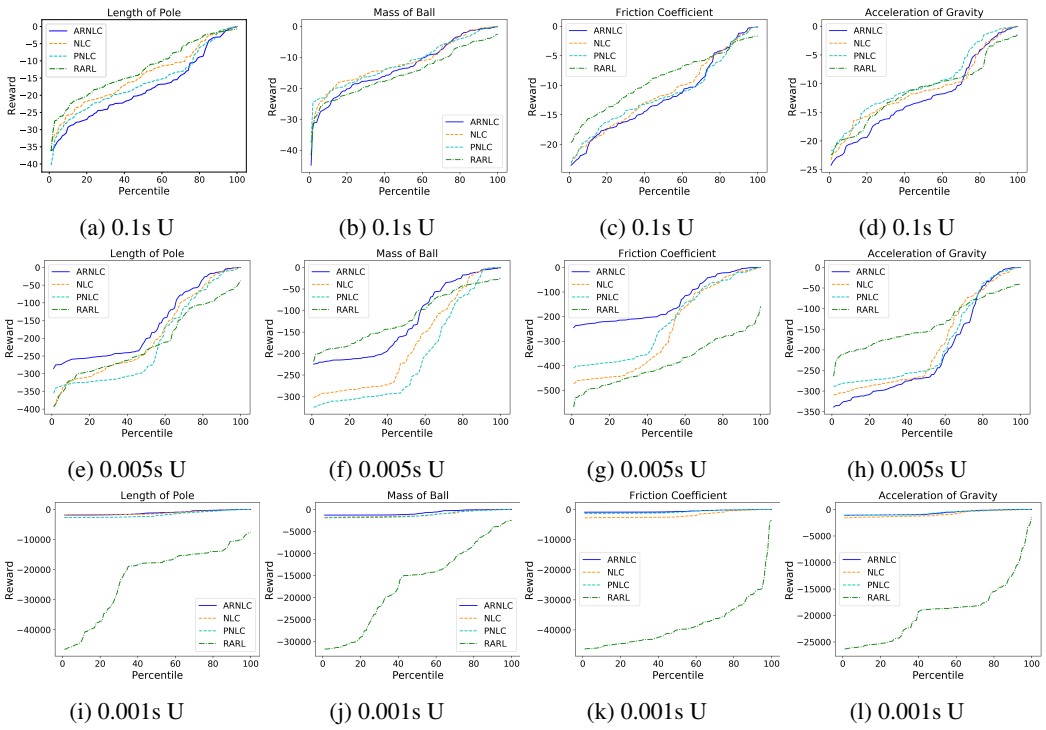

Figure A-13: Percentile plots of Pendulum under uniform (U) perturbations in testing with control interval set to 0.1s, 0.005s and 0.001s, respectively.

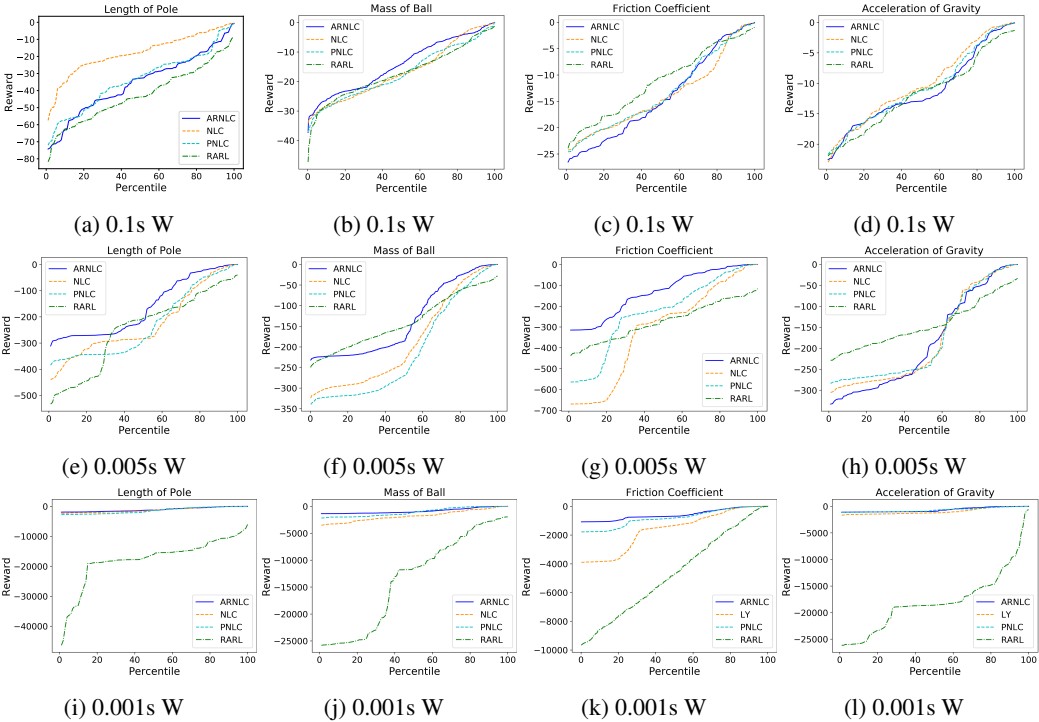

Figure A-14: Percentile plots of Pendulum under learned adversary's worst-case (W) perturbations in testing with control interval set to 0.1s, 0.005s and 0.001s, respectively.

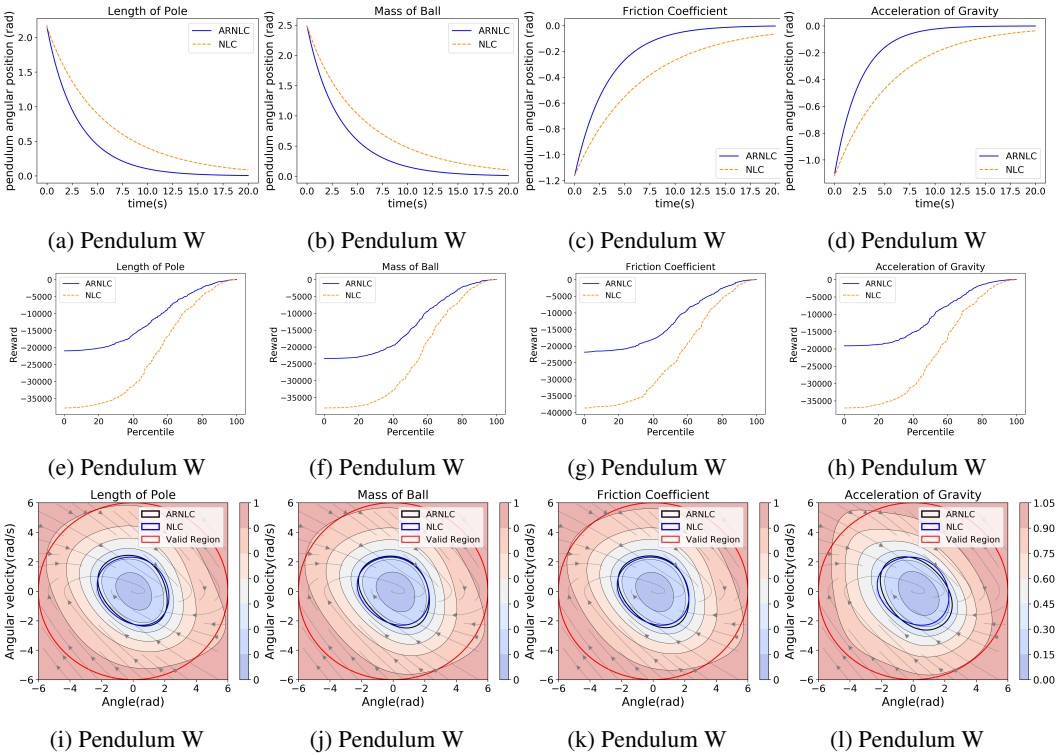

Figure A-15: Control curves, percentile plots and regions of attraction for pendulum balancing in continuous-time control under learned adversary's worst-case (W) perturbations in testing.

## A.5 ADDITIONAL EXPERIMENTAL RESULTS WITH UNKNOWN SYSTEM DYNAMICS SCENARIOS

This section reports experimental results for unknown system dynamics. We carry out the same experiments on all the tasks, while assuming that we have no access to the system dynamics. Therefore, we utilize Algorithm 1 in the main text in Pendulum, Cart Pole, Car Trajectory Tracking and 2-link Pendulum tasks. The evaluation of robust MPC is excluded here, since it requires to know exactly the system dynamics, which makes it no longer feasible.

### A.5.1 CONTROL OF PERTURBED NONLINEAR SYSTEMS

We run the training process of learning-based algorithms on Pendulum, Cart Pole, Car Trajectory Tracking, and 2-link Pendulum utilizing Algorithm 1 in the main text until convergence. Control curves under uniform (U) perturbations are shown in Fig. A-16, while control curves under worst-case (W) perturbations learned by the adversary are illustrated in Fig. A-17. We use the learned adversary in RARL to add perturbations in worst-case testing. This is because for unknown system dynamics, the adversary's actions are the additive error to the output of the learned environment model, and in testing, we require the perturbations in the true dynamics (e.g., physical parameters or external force). We observe that our ARNLC can achieve asymptotic stability under both test scenarios in almost all the tasks, while it reaches the stability the fastest compared to the other baselines. The learned adversary's external force perturbations in the Car Trajectory Tracking task is the only test scenario where our ARNLC fails to reach the equilibrium point, as shown in Fig. A-17(i). Though NLC reaches the stability in some tasks, it fails to reach the equilibrium point in Pendulum W, Car Tracking W and 2-link Pendulum U and W, when the perturbation type in testing is the external force. PNLC trained under uniform sampled perturbations outperforms NLC in some tasks (e.g., Pendulum and Cart Pole), but is worse in Car Tracking W and 2-link Pendulum U.

We additionally compute the percentile of rewards for controller policies achieved by each algorithm to further evaluate their robustness. We run each policy with 100 different initial system states in

each perturbed task, and then sort the cumulative negative adversary reward of each run to obtain the $n$-th percentile. The results obtained under uniform (U) perturbations are shown in Fig. A-18, while percentile plots under worst-case (W) perturbations learned by the adversary are illustrated in Fig. A-19. In general, control policies that can gain higher rewards at the same percentile perform better. While control policies in a lower percentile have better control performance if they receive the same reward, i.e., they can gain higher rewards with more episodes. We observe that our ARNLC can receive the highest rewards under both test scenarios in all the tasks. RARL sometimes fails to reach the stability in Car Trajectory Tracking, and therefore, RARL sometimes receives extremely low rewards in this task.

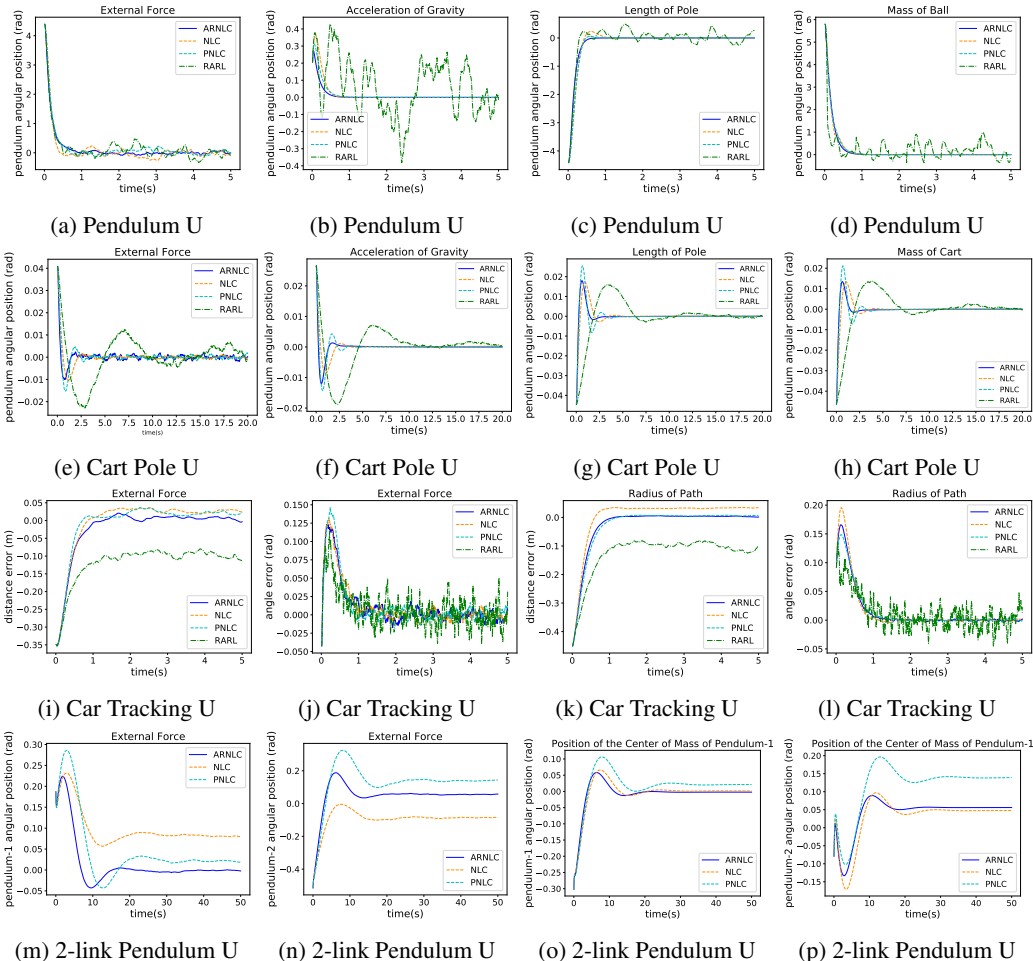

Figure A-16: Control curves of Pendulum, Cart Pole, Car Trajectory Tracking and 2-link Pendulum under uniform (U) perturbations in testing.

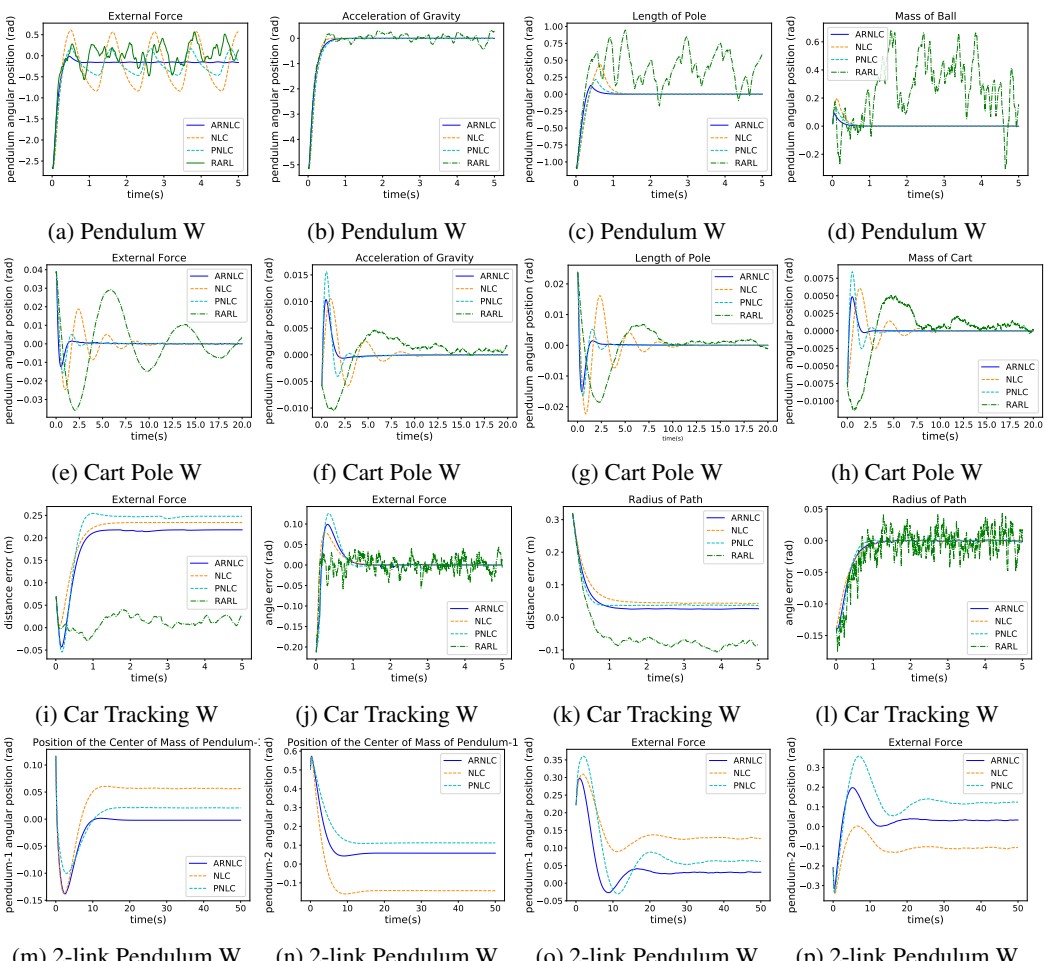

(a) Pendulum W     (b) Pendulum W     (c) Pendulum W     (d) Pendulum W

(e) Cart Pole W     (f) Cart Pole W     (g) Cart Pole W     (h) Cart Pole W

(i) Car Tracking W     (j) Car Tracking W     (k) Car Tracking W     (l) Car Tracking W

(m) 2-link Pendulum W     (n) 2-link Pendulum W     (o) 2-link Pendulum W     (p) 2-link Pendulum W

Figure A-17: Control curves of Pendulum, Cart Pole, Car Trajectory Tracking and 2-link Pendulum under learned adversary's worst-case (W) perturbations in testing.

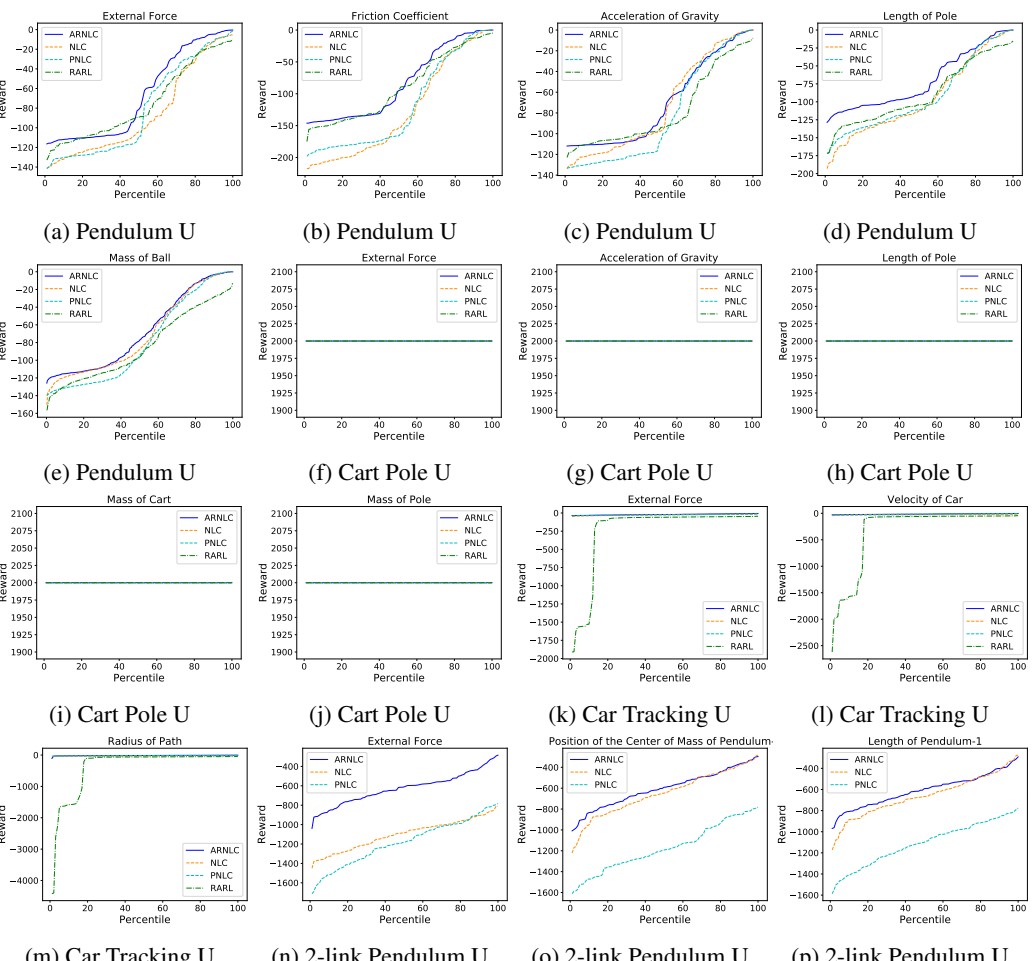

Figure A-18: Percentile plots of Pendulum, Cart Pole, Car Trajectory Tracking and 2-link Pendulum under uniform (U) perturbations in testing.

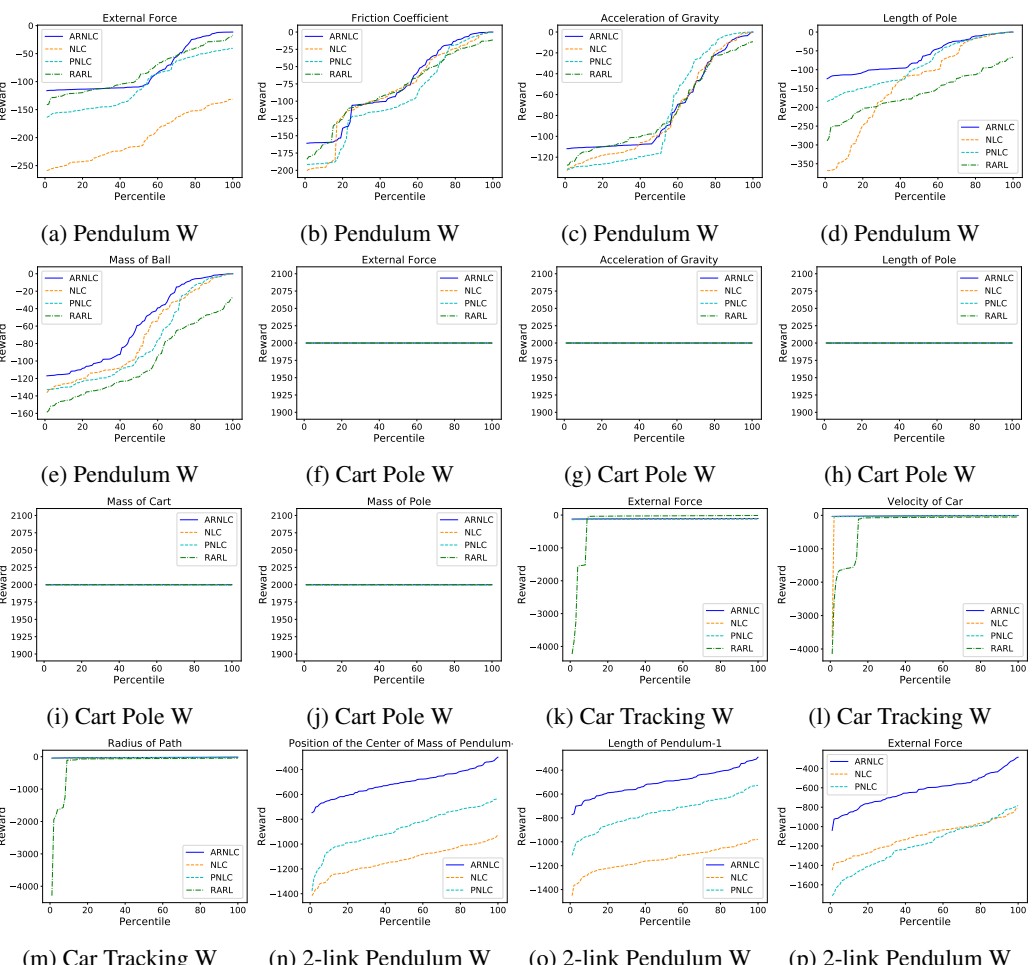

(a) Pendulum W    (b) Pendulum W    (c) Pendulum W    (d) Pendulum W

(e) Pendulum W    (f) Cart Pole W    (g) Cart Pole W    (h) Cart Pole W

(i) Cart Pole W    (j) Cart Pole W    (k) Car Tracking W    (l) Car Tracking W

(m) Car Tracking W    (n) 2-link Pendulum W    (o) 2-link Pendulum W    (p) 2-link Pendulum W

Figure A-19: Percentile plots of Pendulum, Cart Pole, Car Trajectory Tracking and 2-link Pendulum under learned adversary's worst-case (W) perturbations in testing.

### A.5.2 GENERALIZATION IN PERTURBATION SPACE

In this subsection, we evaluate the generalization capability of learning-based algorithms in the entire perturbation space. We observe that ARNLC achieves the best generalization performance under most unknown system dynamics and among all the combinations of different physical parameters. RARL presents the worst performance in all the tasks except for Pendulum.

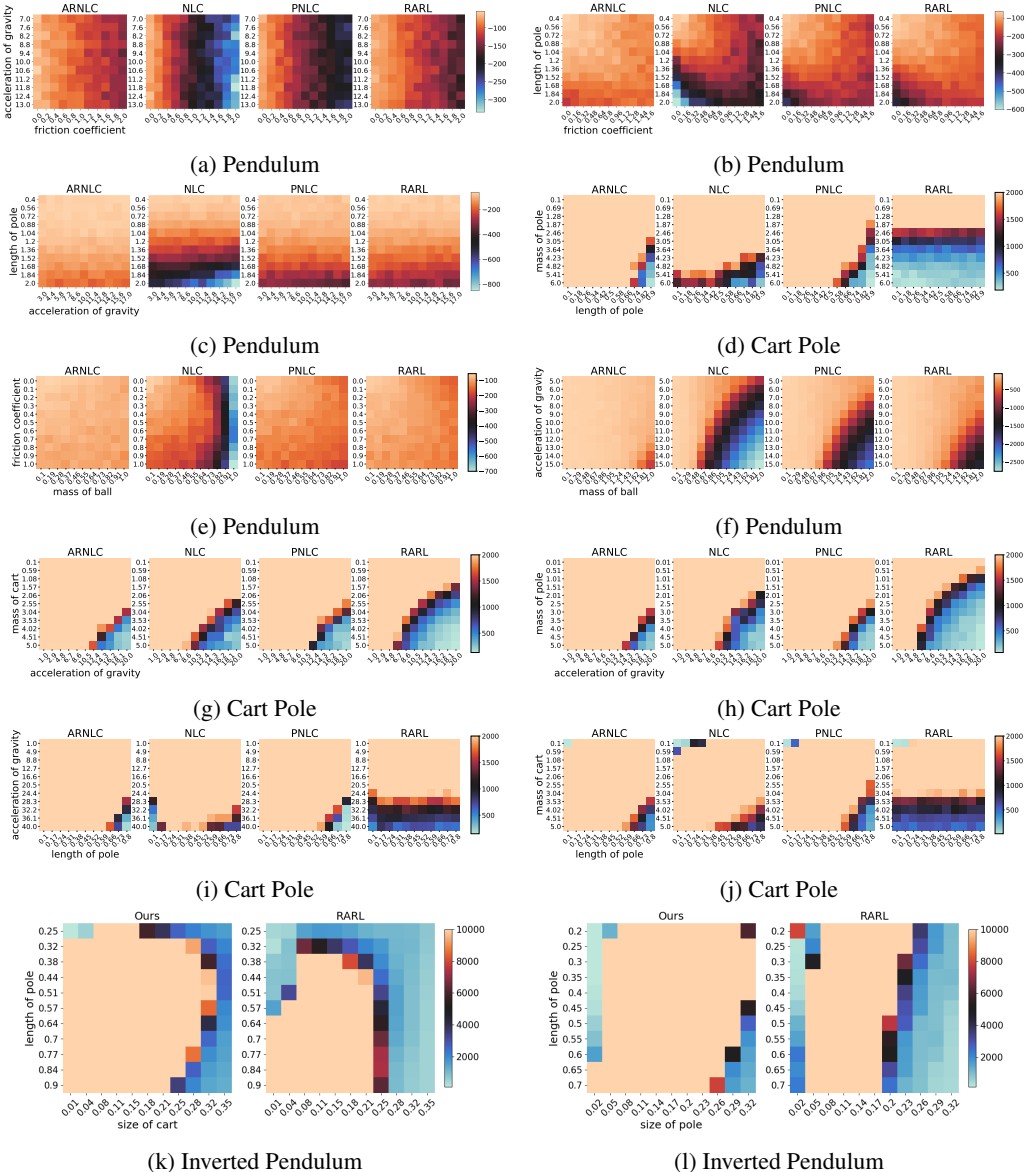

Figure A-20: Heatmap of averaged controller's reward for Pendulum, Cart Pole and Inverted Pendulum.

### A.5.3 IMPACT OF CONTROL INTERVALS

We evaluate the impact of different control intervals (0.01s, 0.1s, 0.005s and 0.001s) to our ARNLC on Pendulum with perturbation types (Length of Pole, Mass of Ball, Friction Coefficient, Acceleration of Gravity) different from the results presented in the main text. The resulting control curves obtained for Pendulum under uniform perturbations are shown in Figs. A-16(a)-A-16(d) and Fig. A-21. And the control curves under perturbation of learned adversary are shown in Figs. A-17(a)-A-17(d) and Fig. A-22. The resulting percentile plots obtained for Pendulum under uniform perturbations are shown in Figs. A-18(a)-A-18(e) and Fig. A-23. And the percentile plots under perturbation of learned adversary are shown in Figs. A-19(a)-A-19(e) and Fig. A-24. The results demonstrate that our ARNLC is robust to different control intervals.

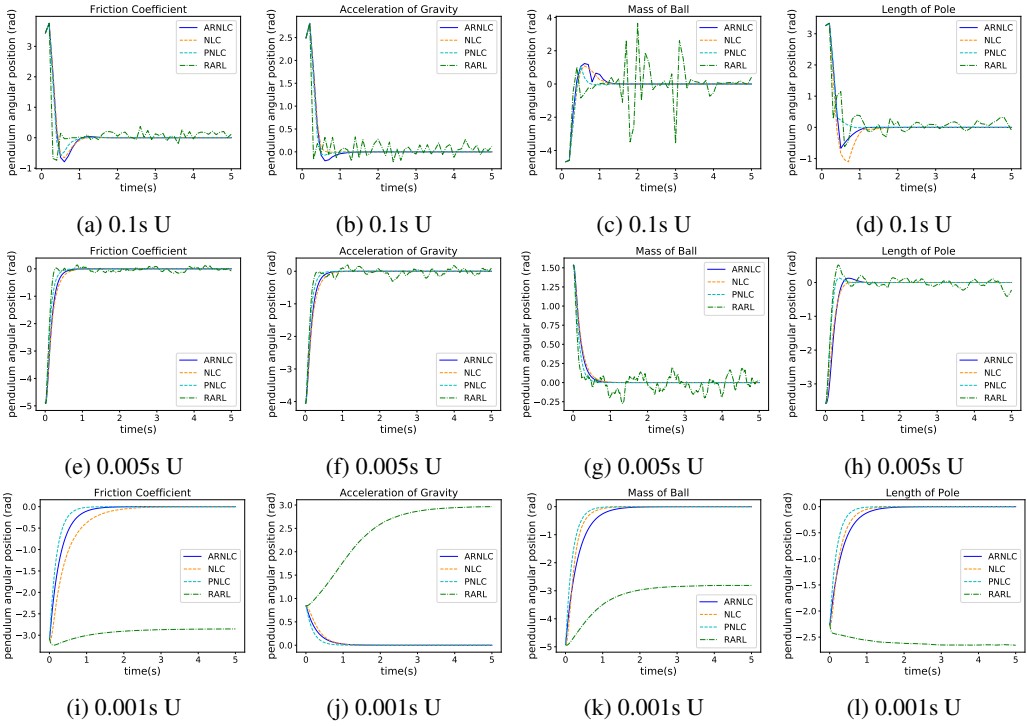

Figure A-21: Control curves of Pendulum under uniform (U) perturbations in testing with control interval set to 0.1s, 0.005s and 0.001s, respectively.

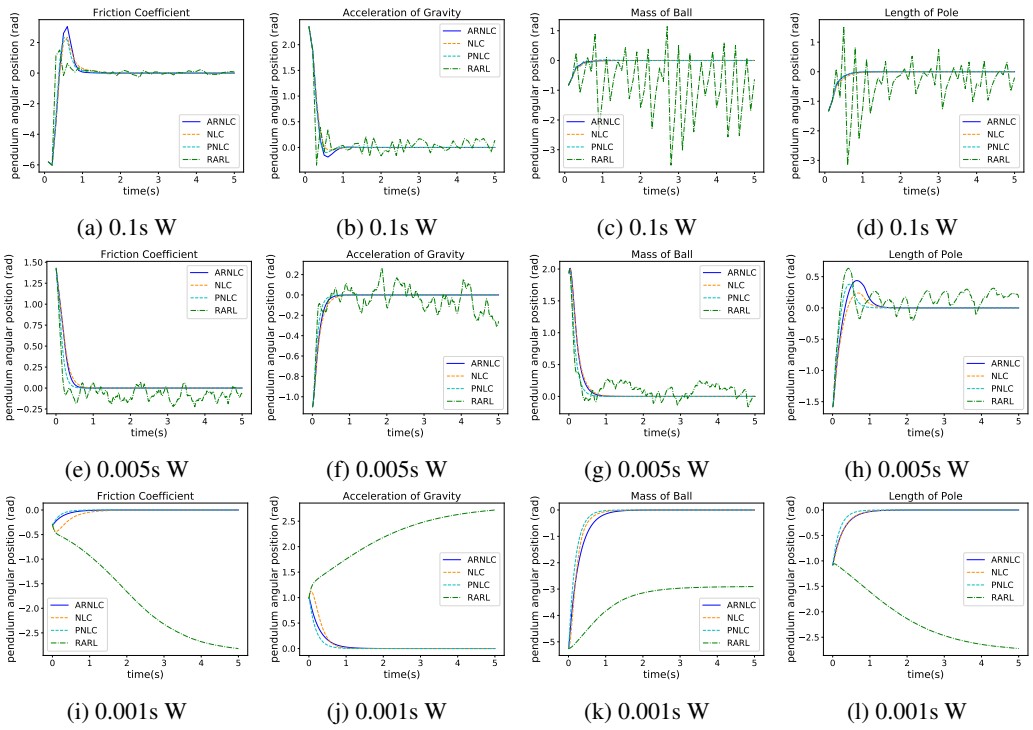

Figure A-22: Control curves of Pendulum under learned adversary's worst-case (W) perturbations in testing with control interval set to 0.1s, 0.005s and 0.001s, respectively.

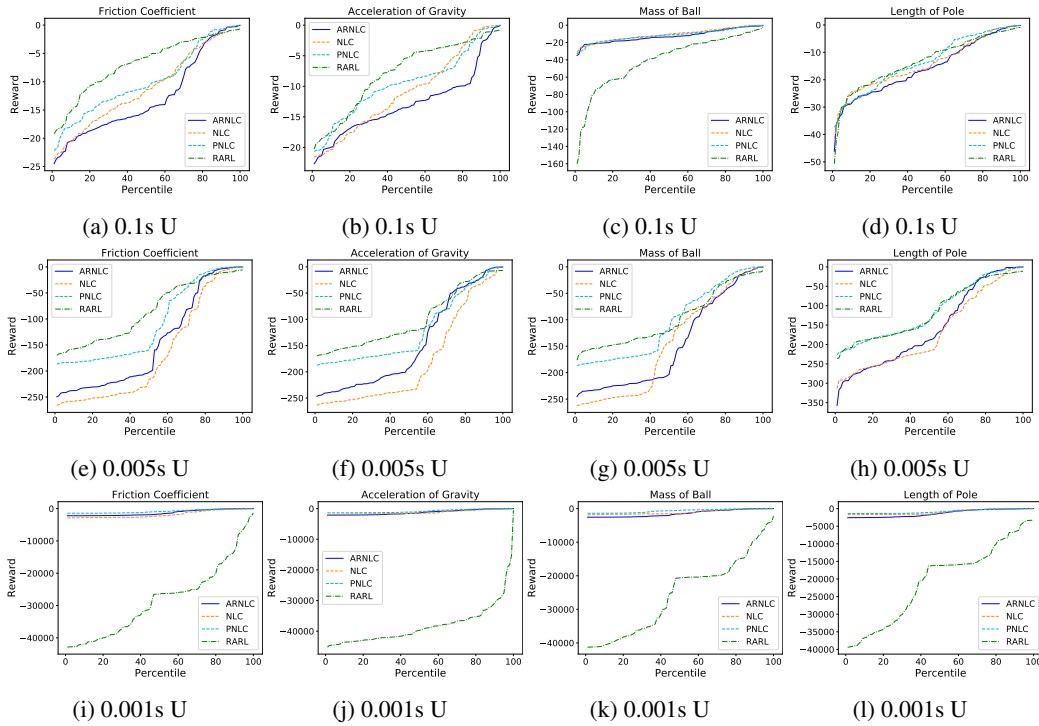

Figure A-23: Percentile plots of Pendulum under uniform (U) perturbations in testing with control interval set to 0.1s, 0.005s and 0.001s, respectively.

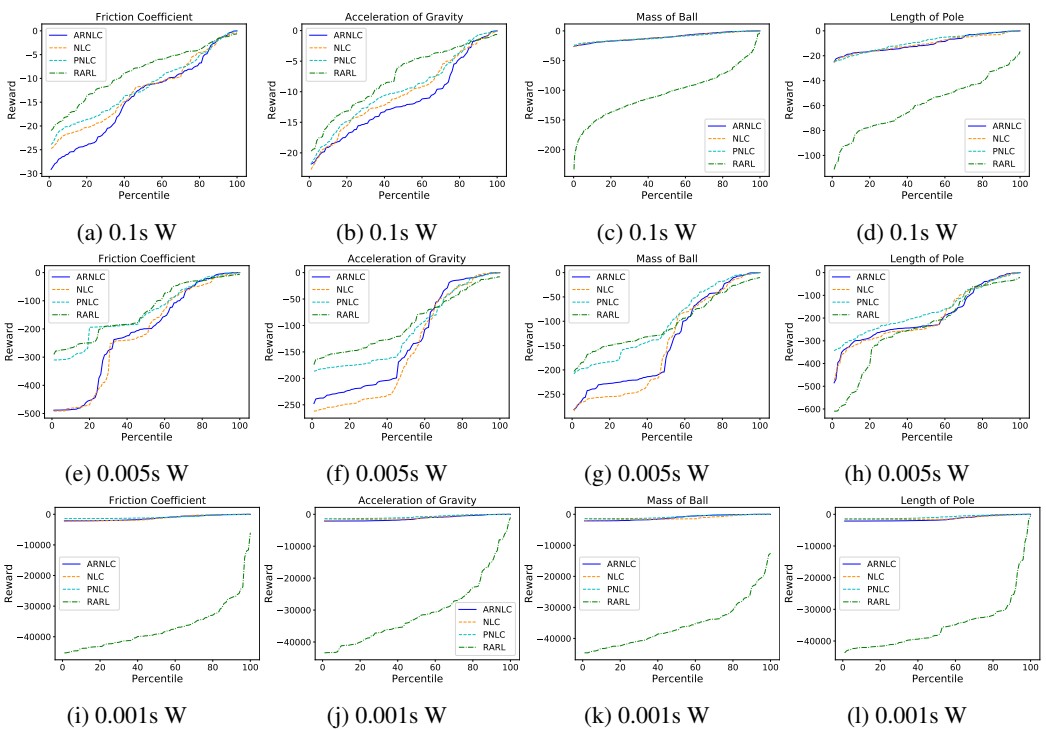

Figure A-24: Percentile plots of Pendulum under learned adversary's worst-case (W) perturbations in testing with control interval set to 0.1s, 0.005s and 0.001s, respectively.

