# OpenReview forum: "Adversarially Robust Neural Lyapunov Control"
_ICLR.cc/2023/Conference — Submitted to ICLR 2023_

### Official Review · Reviewer_tGgb · 2022-10-24

**Confidence:** 4
**Clarity, Quality, Novelty And Reproducibility:** See above.
**Correctness:** 3
**Technical Novelty And Significance:** 2
**Empirical Novelty And Significance:** 2
**Recommendation:** 5

**Strength And Weaknesses:**

Strength:
* The writing is clear and well-organized.
* Detailed Description of the experiment setup and results

Weaknesses:
* The intellectual contribution of this proposed work in comparison to existing neural Lyapunov control seems incremental. The basic framework is based on the Neural Lyapunov Control paper; with the same Lyapunov risk function, the same falsifier, and the same training process, with the addition of state transition dynamics uncertainty.

* In addition, the performance gain of the algorithm compared to Neural Lyapunov Control seems not obvious. For instance, in Fig A-15, the region of attraction of the proposed approach (ARNLC) looks similar to the Neural Lyapunov control (NLC) method. As shown in the Neural Lyapunov control paper, a better Lyapunov function usually leads to a large ROA - so is the learned Lyapunov better than the NLC without considering the disturbances in dynamics?

* On the same vine, not enough visualization about the learned Lyapunov function is provided (like Fig 2 in the Neural Lyapunov Control paper). I am interested in how the perturbation influences the Lyapunov functions. With the perturbation, some states x which satisfy the Lyapunov condition can have negative V(x) or positive V(x)’ when the perturbation is large, it might demonstrate some interesting properties. In particular, a comparison of the learned Lyapunov function shape for ARNLC in comparison to the learned Lyapunov function without adversarial training (NLC) would be interesting, at least for the pendulum example - and comment on the difference.

* Adversarial training usually requires large samples and works for small perturbations. However, in the proposed method, the authors consider the disturbances in the system model. For example, in the Pendulum example, a change of the ball mass, pole length, friction coefficient, and gravity, etc. But then the model disturbances are converted to a state perturbation to be considered in the training, i.e., xk'=M(x_k, a_k)+a_k^v. How do you convert the model discrepancy to a value of a_k^v? What range for the disturbance $a_k^v$ is reasonable to be considered? Also, in many nonlinear control problems, a small change in the system parameters can lead to a significant difference in the state transition, which makes a_k^v sufficiently large.

* Can the authors better describe the difference between PNLC and ARNLC? There seems no proper description of PNLC.

* For figure 1, for example, in (e) and (f), in (e), the curve is increasing, and in (f) the curve is decreasing, it is hard to compare. And some plots do not share the same y limits, which makes the results hard to follow.

* For the training process in the pseudo-code, xk'=M(x_k, a_k)+a_k^v, where M is the learned model, despite that this learned model will introduce error. In the training process, at least for the forward pass, why not use the real simulator to generate data (e.g., like what the authors did in "Sablas: Learning Safe Control for Black-Box Dynamical Systems"? Using the learned model may bring unnecessary errors.


**Summary Of The Paper:**

This paper considers jointly learning a stable controller and neural network represented by the Lyapunov function, which is "empirically" robust with respect to the dynamics difference between the training and test environment. Experiments are conducted on Pendulum, Cart Pole, car tracking, and 2-link pendulum.

**Summary Of The Review:**

This paper considers jointly learning a stable controller and neural network represented by the Lyapunov function, which is "empirically" robust with respect to the dynamics difference between the training and test environment. Experiments are conducted on Pendulum, Cart Pole, car tracking, and 2-link pendulum. Main concern lies in the significance of the proposed approach both intellectually and empirically.

---

> ### Author Response · Authors · 2022-11-19
> **Response to Reviewer tGgb**
>
> We would like to thank the reviewer for the detailed comments. Here, we would like to provide our response to these comments.
>
> **Comment 1**: “The intellectual contribution of this proposed work in comparison to existing neural Lyapunov control seems incremental. The basic framework is based on the Neural Lyapunov Control paper; with the same Lyapunov risk function, the same falsifier, and the same training process, with the addition of state transition dynamics uncertainty.”
>
> **Response**: We newly design the perturbed Lyapunov risk for learning the controller, which takes the adversary’s perturbations into consideration. Also, we formulate an optimization problem for adversarially robust controller learning, introducing an adversary into an alternate training procedure.
>
> **Comment 2**: “For instance, in Fig A-15, the region of attraction of the proposed approach (ARNLC) looks similar to the Neural Lyapunov control (NLC) method. As shown in the Neural Lyapunov control paper, a better Lyapunov function usually leads to a large ROA - so is the learned Lyapunov better than the NLC without considering the disturbances in dynamics?”
>
> **Response**: The proposed ARNLC aims to acquire a policy robust to perturbation. Hence a Lyapunov function that leads to a similar ROA under perturbation as the one without perturbation (NLC) is expected instead of a Lyapunov function of larger ROA.
>
> **Comment 3**: “not enough visualization about the learned Lyapunov function is provided (like Fig 2 in the Neural Lyapunov Control paper). I am interested in how the perturbation influences the Lyapunov functions.”
>
> **Response**: Thanks for the suggestion. We have provided visualization about the learned Lyapunov function in Section 2 in the supplementary material. The Lyapunov function learned by our ARNLC and the original NLC is similar, and the contour line, which denotes the region of attraction (ROA), is also similar. Hence, our proposed ARNLC acquires a policy robust to perturbation.
>
> **Comment 4**: “What range for the disturbance $a_k^v$ is reasonable to be considered?”
>
> **Response**: We have conducted experiments on the comparison of different perturbation ranges. The results have been provided in Section 4 in the supplementary material, which present that the perturbation range we tune and choose can strike the trade-off between stable training and robust controlling.
>
> **Comment 5**: “Can the authors better describe the difference between PNLC and ARNLC? There seems no proper description of PNLC.”
>
> **Response**: The perturbation applied in the dynamical system in PNLC is from a uniformly random adversary. While in ARNLC, we learn an optimal RL-trained adversary to impose the disturbance in the dynamical system. Therefore, the difference between them lies in that PNLC randomly samples from a uniform probability distribution and treats it as perturbation, while ARNLC uses the RL-trained adversary to choose the optimal perturbation in the face of each state.
>
> **Comment 6**: “For figure 1, for example, in (e) and (f), in (e), the curve is increasing, and in (f) the curve is decreasing, it is hard to compare. And some plots do not share the same y limits, which makes the results hard to follow.”
>
> **Response**: When conducting test, the initial state of each task in the environment is uniformly randomly sampled. The reason why some curves are increasing and some curves are decreasing lies in that some initial states are positive, while some initial states are negative. However, our ARNLC all converges to zero, which denotes that our ARNLC can achieve the asymptotic stability.
>
> **Comment 7**: “For the training process in the pseudo-code, $x_{k'}=M(x_k, a_k)+a_k^v$, where $M$ is the learned model, despite that this learned model will introduce error. In the training process, at least for the forward pass, why not use the real simulator to generate data?”
>
> **Response**: Thanks for the suggestion. We assume that we have no access to a well-designed simulator.

---

### Official Review · Reviewer_g1he · 2022-10-28

**Confidence:** 5
**Correctness:** 3
**Technical Novelty And Significance:** 4
**Empirical Novelty And Significance:** 3
**Recommendation:** 5

**Clarity, Quality, Novelty And Reproducibility:**

I am overall excited about the topic this paper is studying. The paper is generally clear and easy to follow and writing quality is good, although some key claims seem
to be problematic. Novelty is good although there are issues in claims and approaches,
as discussed in "questions and weakness" above.

**Strength And Weaknesses:**

Strengths:

1. The paper studies an important problem of learning stable and robust neural
network controllers. The approach of combining a RL agent as the adversarial
with neural Lyapunov learning is interesting. The use of neural Lyapunov under
the context of adversarial robustness is also novel.

2. Empirical results are comprehensive and the visualization of results look
nice. The proposed approach achieves empirical stability in a few simulation
runs and it converges better under the attack of the learned adversary.

Questions and weaknesses:

One major benefit of using Lyapunov theory in neural controllers is that it can
formally guarantee stability. However, the paper does not convincingly show
such a formal guarantee.  The paper emphasized a few times that the learned
policy enjoys a theoretical guarantee of stability (e.g., in the abstract
"learned control policy enjoys a theoretical guarantee of stability"), however
most evaluations were done only empirically, and there is no real stability
guarantee. The benefits of neural Lyapunov control is that it can provide
stability guarantee, however training with an empirical Lyapunov loss function
with an RL learned non-optimal adversaray based on approximated system dynamics
seem far away from giving formal guarantees. The algorithm presented in
Algorithm 1 has not convergence guarantees, and the paper is also unclear about
the details of the falsifier (which might be able to give some formal
guarantees) used during training.

The biggest question is, given a fixed system dynamic function and a fixed
controller function, can you *formally* show that the controller is stable
under *any bounded adversary* inside a certain region of convergence using Lyapunov theory?  The
results presented in this paper do not support this - results show a few
empirical simulation of the systems only.

If the controllers do have guaranteed stability, then these critical results
must be presented:

1. How large is the region of the attraction?
2. How large is the allowed perturbation for the adversary for the guarantee to hold?
Technically, the adversary needs to be the optimal adversary to give a meaningful
theoretical guarantee. If the guarantee is only valid for a trained adversary it is much
weaker since we don't know how good the adversary is.
3. What is the cost of computing such a guarantee (e.g., time to run the falsifier)?

If these guarantees cannot be obtained, then this paper needs a major revision to
remove claims on stability guarantees, such as those in abstract. And it will
become less exciting since the use of Lyapunov theory does not provide any
formal guarantees. In that case, it is unclear if the use of Lyapunov theory is
necessary, and a strong baseline would be using normal RL training to replace
the Lyapunov loss to guarantee empirical stability. The benefits of using
Lyapunov is unclear here. In addition, since the proposed approach is evaluated
under the learned adversary, there is a chance that the controller overfits to this
adversary so the evaluation is not fair - this must be addressed in experiments.


**Summary Of The Paper:**

The paper studies training a neural network based controller under potential
adversaries that maliciously alter the states of the dynamic system. The
technical approaches the paper uses consist of three parts: (1) train an
approximate environment model for the system; (2) use typical reinforcement
learning algorithm like PPO to train an adversarial on the approximated system
model to find the worst case perturbation; and (3) train a neural network
controller under this adversary using Lyapunov loss. The paper claims to learn
control policy with theoretical guarantee of stability using this approach.
Experimentally, the paper evaluates this approaches on a few classic control
environments such as pendulum and 2-link pendulum.

**Summary Of The Review:**

I feel this paper definitely studies an intriguing problem and I got excited
when seeing its title and abstract. Unfortunately I feel some key claims in the
paper are not well supported and the paper is not able to deliver its promise.
So I tend to reject the current version of this paper, but I am happy to
discuss with the authors further for clarifications. I am glad to re-evaluate
the paper based on responses from the authors.

---

> ### Author Response · Authors · 2022-11-19
> **Response to Reviewer g1he**
>
> We would like to thank the reviewer for the detailed comments. Here, we would like to provide our response to these comments.
>
> **Comment 1**: “The biggest question is, given a fixed system dynamic function and a fixed controller function, can you formally show that the controller is stable under any bounded adversary inside a certain region of convergence using Lyapunov theory?”
>
> **Response**: Thanks for this suggestion. We are going to give derivation of theoretical guarantee of stability in the future version.
>
> **Comment 2**: “In addition, since the proposed approach is evaluated under the learned adversary, there is a chance that the controller overfits to this adversary so the evaluation is not fair - this must be addressed in experiments.”
>
> **Response**: Thanks for the suggestion. We have conducted evaluation under a newly trained adversary. The results have been provided in Section 1 in the supplementary material. Our ARNLC still holds the advantage of reaching the stability the fastest compared to the other baselines under a newly trained adversary's perturbation.

---

### Official Review · Reviewer_6NKw · 2022-11-02

**Confidence:** 4
**Correctness:** 3
**Technical Novelty And Significance:** 2
**Empirical Novelty And Significance:** 3
**Recommendation:** 5

**Clarity, Quality, Novelty And Reproducibility:**

The paper seems to be written clear/well and it contains valuable novel results. I'm not convinced by authors' claims regarding the "theoretical guarantee" for their proposed ARNLC method, but the proposed method and the empirical evaluations is valuable.

**Strength And Weaknesses:**

Strength:

[1] The LNC robustness stabilization problem studied here is interesting and important, from both theory and application perspective.
[2] Overall speaking, the paper presentation is clear and I found it straightforward to follow.
[3] The empirical evaluation results based on several case studies are helpful to demonstrate the advantages of the proposed method, in comparison with other available baseline methods in the literature.

Weakness:
[1] In the paper abstract, it is claimed by authors that, "the learned control policy enjoys a theoretical guarantee of stability". However, based on my reading of the paper, I didn't see any rigorous theorem & proofs about such "theoretical guarantee of stability", for the proposed ARNLC algorithm. What's presented in the paper is some "empirical evaluation based on cases studies" plus some non-rigorous comment/explanations of the proposed algorithms. So the "theoretical guarantee of stability" seems to be a over-claim here and it is inappropriate. Note that, the Theorem 1 about using stability guarantee with Lyapunov function was only for using the exact control policy and Lyapunov function without approximation, and for the proposed ARNLC method with multiple approximations (and could potentially violate Eq2 conditions many times), there is not any rigorous "theoretical guarantee of stability" any more. In my opinion, lacking of any rigorous theoretical performance analysis of the proposed ARNLC method (which has multiple approximations) is a main weakness of this paper, and the paper would be significantly improved if the authors are able to add/derive any rigorous stabilization performance analysis of the proposed ARNLC method. And at the very least, the authors should remove their claim about  "the learned control policy enjoys a theoretical guarantee of stability" for their proposed ARNLC method.

[2] The system considered in this paper is general continuous-time nonlinear dynamic systems, without other assumptions/conditions (e.g., lipschitz condition on f) on the system dynamics. Is it really the case that the proposed ARNLC method could work for such "general nonlinear systems" with "theoretical guarantees about the stability", without any other assumptions/conditions on system dynamics? It is well known in nonlinear dynamic system control theory that, for nonlinear dynamic systems, it could even possibly has a nonlinear system finite escape time. For the proposed ARNLC method with multiple approximations,  I'm not convinced that the closed-loop system using the proposed ARNLC method could achieve  "theoretical guarantee of asymptotical stability" for such general nonlinear unknown dynamic systems (without other assumptions/conditions). Rigorous mathematical analysis about the closed loop system performance for the proposed ARNLC method would be required if the authors want to justify their claim about theoretical stability guarantee of their proposed method. Without that rigorous mathematical analysis, the results in this paper is still valuable/helpful, but its main contributions is from empirical application side, not theoretical side.

[3] In page 4, it was mentioned that Equation (7) is an empirical unbiased estimator of Equation (6). But I think Equation (7) is an empirical unbiased estimator of Equation (5) rather than Equation (6)? Also, for the notation N in equation (7), it might be good to add some clarification for it, as well as what value of N was used for the approximation and also justify it.

[4] In the paper, it was mentioned that "if we can train a controller under the worst-case perturbation (which degrades the performance of its policy to the most) in a certain range, the controller then obtains a conservative policy that is robust to any perturbation within that same range". In my opinion, such a statement is inaccurate and also not sufficiently convincing, and it might possibly cause misunderstanding about the proposed method's performance against generic perturbations within that same range. In particular, considering all the perturbations within that same range, the controller was trained/optimized against the so-called "worst-case" perturbation and thus the final obtained controller might possibly perform better against that specific perturbation (since it has be somewhat trained/optimized to adapt to that type of perturbations),  and there is not any rigorous mathematical analysis to support the statement that the obtained controller would perform well for all other perturbations (for which the obtained controlled were not trained/optimized against) within that same range. It might be helpful for the authors to explain/elaborate this in a more accurate/convincing way to avoid potential misunderstandings about the proposed method.


**Summary Of The Paper:**

In this paper, the authors propose an adversarially robust neural Lyapunov control (ARNLC) method to improve the robustness and generalization capabilities for Lyapunov theory-based stability control. They claimed that the main contributions are:

[1] propose a perturbed Lyapunov risk for learning the control policy under perturbations.

[2] formulate an optimization problem for adversarially robust controller learning, to learn a policy in face of the worst-case perturbations that are imposed by the RL-trained adversary.

[3] propose an adversarially robust neural Lyapunov control (ARNLC) approach to approximately solve this problem, and demonstrate its performance on several stability control tasks.

**Summary Of The Review:**

This paper studied an interesting problem and proposed a novel ARNLC method for robust stabilization control problem of general nonlinear dynamic systems, and it contains some novel valuable results. The advantages of the proposed method are demonstrated via multiple empirical evaluation case studies. The main weakness is lacking of theoretical analysis of the proposed ARNLC method, and I'm not convinced by the author's claim about "the learned control policy enjoys a theoretical guarantee of stability".

I think the contribution of this paper from theoretical perspective is weak, but the proposed method and empirical evaluations are still valuable. The paper contribution would be much more significant if the authors are able to add rigorous theoretical analysis to prove that "the learned control policy indeed achieves theoretical guarantee of stability for the closed-loop system using their proposed ARNLC method". Without adding that theoretical part, I think this paper's contribution is board-line, and slightly below the acceptance threshold.

---

> ### Author Response · Authors · 2022-11-19
> **Response to Reviewer 6NKw**
>
> We would like to thank the reviewer for the detailed comments. Here, we would like to provide our response to these comments.
>
> **Comment 1**: “In my opinion, lacking of any rigorous theoretical performance analysis of the proposed ARNLC method (which has multiple approximations) is a main weakness of this paper, and the paper would be significantly improved if the authors are able to add/derive any rigorous stabilization performance analysis of the proposed ARNLC method.”
>
> **Response**: Thanks for this suggestion. We are going to give derivation of theoretical guarantee of stability in the future version.
>
> **Comment 2**: “But I think Equation (7) is an empirical unbiased estimator of Equation (5) rather than Equation (6)?”
>
> **Response**: Thanks for the careful reading. We apologize for this typo, and will correct it in the future version.
>
> **Comment 3**: “In particular, considering all the perturbations within that same range, the controller was trained/optimized against the so-called "worst-case" perturbation and thus the final obtained controller might possibly perform better against that specific perturbation (since it has be somewhat trained/optimized to adapt to that type of perturbations), and there is not any rigorous mathematical analysis to support the statement that the obtained controller would perform well for all other perturbations (for which the obtained controlled were not trained/optimized against) within that same range.”
>
> **Response**: Sorry for the unclear statement. The worst-case perturbation here is the worst-case among the same range and type as the ones during controller training.

---

> > ### Comment · Reviewer_6NKw · 2022-12-12
> > **Responses to Authors' response**
> >
> > I've checked out all other reviewers' comments as well as the authors' responses. I'd keep my rating unchanged and I think it is marginally below the acceptance threshold.

---

### Official Review · Reviewer_1FTL · 2022-11-05

**Confidence:** 4
**Correctness:** 2
**Technical Novelty And Significance:** 2
**Empirical Novelty And Significance:** 2
**Recommendation:** 3

**Clarity, Quality, Novelty And Reproducibility:**

The paper is overall well written and clear. As a minor note, the papers notation for dynamical systems is rather confusing. Instead of using $x_t$ for the state in continuous time, most works in both RL and control use $x(t)$ for states in continuous time and $x_t$ for states in discrete time. This becomes particularly important when the authors train the system using PPO since they use an Euler discretization of the continuous dynamics and thus have a series of discrete states and actions.

**Strength And Weaknesses:**

# Strengths
1. Thorough description of experimental setup and methodology.
2. Experiments detailed sufficiently to be reproducible even without code.

# Weaknesses
1. ***Lack of Stability Guarantees***:
In the abstract the authors claim: "the learned control policy enjoys a theoretical guarantee of stability."
Although this may be true for NLCs that achieve zero Lyapunov risk loss, ARNLC significantly changes the setting by the inclusion of adversarial disturbances where it is not clear how the guarantees go through. To make this claim the authors should include a theorem with the specific assumptions on the adversary or modeling error.
4. ***Lack of Reference to robust control theory***:
The topic of robustness with Lyapunov theory has been extensively studied not just in controls but also in the area of controls applied to learning. At least some comparison should be done with standard robust control approaches for this to be accepted. Using black-box environments should not be an obstacle given that the authors are learning an approximation of the dynamics in $\mathcal{M}_\eta$. It may be that control methods achieve similar performance without having to learn an adversary or policy. Possible papers to cite include but are not limited to (I highlight these references because they discuss robustness with provable guarantees in the learning context because stability guarantees are mentioned in the abstract):
    1. Robustness in the context of Lyapunov theory precisely when trying to use learning to overcome model mismatch: Taylor, A. J., Dorobantu, V. D., Krishnamoorthy, M., Le, H. M., Yue, Y., & Ames, A. D. (2019, December). A control lyapunov perspective on episodic learning via projection to state stability.
   2. Robust  MPC with provable guarantees: Aswani, A., Gonzalez, H., Sastry, S. S., & Tomlin, C. (2013). Provably safe and robust learning-based model predictive control. Automatica, 49(5), 1216-1226.
1. ***Unclear theoretical setting***:
The authors should clearly specify the assumptions on the dynamics function $f$ in particular they should clearly state the assumptions required for uniqueness and existence of solutions on which the Lyapunov stability theorem relies.
1. ***On the generality of control-affine systems*** In the second paragraph the author states that : " However, this approach only considers  the control-affine dynamical systems, not the more general nonlinear one"
A general nonlinear system can always be placed into control affine form by choice of an integral controller. Furthermore the test systems are mechanical systems which can be placed into control-affine form. It would be interesting to see a case where the generality of ARNLC is required.
1. ***On Exponential Stability and Robustness***: It can be shown that exponentially stable systems converge to a small area around the equilibrium point rather than the equilibrium point itself under perturbation. This property is commonly referred to as Input-to-State Stability(ISS). Depending on the magnitude of the perturbation from the nominal dynamics, an exponentially stable system may converge close enough to the equilibrium to achieve the results shown. Some experiments showing that the resulting stability cannot be achieved by a sufficiently exponentially stable controller would strengthen the claims of the paper. I would consider citing the following: Sontag, Eduardo D. "Input to state stability: Basic concepts and results." Nonlinear and optimal control theory. Springer, Berlin, Heidelberg, 2008. 163-220 . or Liberzon, D., Sontag, E. D., & Wang, Y. (2002). Universal construction of feedback laws achieving ISS and integral-ISS disturbance attenuation. Systems & Control Letters, 46(2), 111-127.
1. ***On the generality of additive noise*** Suppose you have a nominal dynamics $f(x,\hat{w})$ and a true dynamics $f(x,w)$. I can write the true dynamics as the nominal dynamics with additive noise as follows: $\dot{x} = f(x,\hat{w}) + \epsilon$ where the additive noise $\epsilon = f(x,w) - f(x,\hat{w})$. The authors mention that some prior work is only restricted to additive disturbances as follows: "However, it is usually restricted to the additive disturbances (Löfberg, 2003) ". The paper would be significantly strengthened by showing a case where an additive disturbance model is not sufficient either theoretically or empirically.
1. ***Unclear performance measurement***:
The paper shows curves of controllers successfully stabilizing the desired input. Although this clearly shows that the controllers can stabilize the system, they may still be wildly inefficient at achieving this goal. You may be able to achieve very high robustness while using very high inputs. To evaluate the performance of the controller, the authors should consider showing the norm of the control input as the pendulum converges. Similar robustness results could be achieved by scaling the control input to increase the scale of the system relative to the noise.
1. ***Unclear Statements about worst-case perturbation***:
> The intuition behind is that if we can train a controller under the worst-case perturbation (which degrades the performance of its policy to the most) in a certain range, the controller then obtains a conservative policy that is robust to any perturbation within that same range.

In general, its not safe to assume that robustness to the worst case perturbation results in robustness to any perturbation. Even in a linear case, making the system robust to a single worst-case perturbation does not stop other perturbations from causing instability. There may be a version of this statement that is true but, again, the authors would have to state the assumptions clearly in a theorem and provide proof.

For the empirical evaluation I am not sure if the ARNLC results are evaluated on a new adversary or the one that was used to train the policy and a lyapunov function. Not retraining the adversary while keeping the weights of the policy constant would seem akin to reporting training accuracies since the policy has been able to adapt to that particular instance of the adversary and may not generalize. An ablation study of a newly trained adversary to which the policy cannot adapt and a uniform random adversary would show empirical evidence of the worst-case perturbation assumption stated previously.

**Summary Of The Paper:**

This work proposes an extension to Neural Lyapunov Control (NLC) to minimize model mismatch: Adversarially Robust Neural Lyapunov Control(ARNLC). This approach treats model mismatch as an additional control input to the system that an adversary may use to reduce the performance of the system. ARNLC then takes a multi-step training procedure. First the authors train on the perturbed Lyapunov Risk function: an extension to the NLC's Lyapunov risk with the new adversary as an input. Then, they train the adversary using PPO in an RL formulation with reward designed to go against the goal of the nominal controller. Finally the authors demonstrate the performance of their approach in synthetic control tasks namely the Pendulum, Cartpole, Car Trajectory Tracking and 2-link pendulum.

**Summary Of The Review:**

Overall this paper tackles the interesting and topical subject of model mismatch and sim-to-real in a fresh way. The use of a trained adversary to in an actor-critic fashion to fit a lyapunov function is certainly novel and has future potential. Unfortunately, this particular instantiation of the idea is lacking a few critical and minor points. Although the experimental section is written clearly, it does not state explicitly that the adversarial policy used for evaluation is new see point [8] above which weakens significantly the empirical claims of the paper. Similarly as stated in point[1] the theoretical guarantees promised do not have a theorem or similar statement in the paper. Finally, the paper does not mention the extensive literature in Robust control on achieving robust stability with Lyapunov-like conditions see [2,5]. Because of these problems I have to recommend the paper for rejection.

---

> ### Author Response · Authors · 2022-11-19
> **Response to Reviewer 1FTL**
>
> We would like to thank the reviewer for the detailed comments. Here, we would like to provide our response to these comments.
>
> **Comment 1**: “To make this claim the authors should include a theorem with the specific assumptions on the adversary or modeling error.”
>
> **Response**: Thanks for this suggestion. We are going to give derivation of theoretical guarantee of stability in the future version.
>
> **Comment 2**: “At least some comparison should be done with standard robust control approaches for this to be accepted.”
>
> **Response**: Thanks for this suggestion. We have compared our ARNLC with robust MPC in the experiment part. As shown in Fig. 1 in the main text, our ARNLC can achieve asymptotic stability while robust MPC fails to achieve asymptotic stability. We will consider adding some citations of robust control approaches in the future version.
>
> **Comment 3**: “The authors should clearly specify the assumptions on the dynamics function in particular they should clearly state the assumptions required for uniqueness and existence of solutions on which the Lyapunov stability theorem relies.”
>
> **Response**: We have conducted experiments on the comparison of different perturbation ranges. The results have been provided in Section 4 in the supplementary material, which show that the perturbation range we tune and choose can strike the trade-off between stable training and robust controlling.
>
> **Comment 4**: “A general nonlinear system can always be placed into control affine form by choice of an integral controller. Furthermore the test systems are mechanical systems which can be placed into control-affine form. It would be interesting to see a case where the generality of ARNLC is required.”
>
> **Response**: The application of ARNLC does not require the careful choice of an integral controller and even the known expression of the system function. It can run on target system and learn the policy from the sampled data.
>
> **Comment 5**: “It can be shown that exponentially stable systems converge to a small area around the equilibrium point rather than the equilibrium point itself under perturbation. This property is commonly referred to as Input-to-State Stability (ISS).”
>
> **Response**: Thanks for this suggestion. We will consider giving the derivation of theoretical guarantee of stability from the perspective of ISS.
>
> **Comment 6**: “The paper would be significantly strengthened by showing a case where an additive disturbance model is not sufficient either theoretically or empirically.”
>
> **Response**: Thanks for this suggestion. We consider the perturbations on environment parameters rather than the additive disturbance. We have empirically evaluated that the performance of the controller trained under the perturbations on environment parameters realized by our ARNLC is better than that trained under the simple additive disturbance realized by PNLC. The results are shown in Figure 1 in the main text. Our ARNLC can achieve asymptotic stability and reach the stability faster than PNLC in all the tasks.
>
> **Comment 7**: “To evaluate the performance of the controller, the authors should consider showing the norm of the control input as the pendulum converges. Similar robustness results could be achieved by scaling the control input to increase the scale of the system relative to the noise.”
>
> **Response**: Thanks for this suggestion. We have shown the norm of the control input in the Section 3 in the supplementary material. Under the disturbance brought by the adversary, our controller input is still in a reasonable range.
>
> **Comment 8**: “it's not safe to assume that robustness to the worst case perturbation results in robustness to any perturbation.”
>
> **Response**: Sorry for the unclear statement. The worst-case perturbation here is the worst-case among the same range and type as the ones during controller training.
>
> **Comment 9**: “An ablation study of a newly trained adversary to which the policy cannot adapt and a uniform random adversary would show empirical evidence of the worst-case perturbation assumption stated previously.”
>
> **Response**: Thanks for the suggestion. We have already conducted evaluation under a uniform random adversary, which is represented as uniform (U) perturbations in the main text. Also, we have conducted evaluation under a newly trained adversary. The results have been provided in Section 1 in the supplementary material. Our ARNLC still holds the advantage of reaching the stability the fastest compared to the other baselines under a newly trained adversary's perturbation.

---

### Decision · Program_Chairs · 2023-01-20

**Decision:**

Reject

**Justification For Why Not Higher Score:**

 The overall approaches in this paper are interesting, but all the reviewers had serious doubts and questions about the theoretical setting and overall approach, to the point that the paper is fairly clearly not yet ready for acceptance.

**Justification For Why Not Lower Score:**

 NA

**Metareview: Summary, Strengths And Weaknesses:**

Thank you for your submission to ICLR.  This paper presents an adversarially robust Lyapunov-based controller that attempts to better account for model discrepancy between training and test distributions. While there do appear to be some interesting aspects to the proposed formulation, the reviewers and I are in agreement (even after author response), that the paper lacks enough connections to existing work in Lyapunov control and is unclear enough in its theoretical setting, that it is not yet ready for publication.